# Technical Note: Representing glacier geometry changes in a semi-distributed hydrological model

Jan Seibert[1], Marc J. P. Vis[1], Irene Kohn[2], Markus Weiler[2], Kerstin Stahl[2]

[1]Department of Geography, University of Zurich, Zurich, 8057, Switzerland
[2]Faculty of Environment and Natural Resources, University Freiburg, Freiburg, 79098, Germany

*Correspondence to* (jan.seibert@geo.uzh.ch)

**Abstract**

Glaciers play an important role in high-mountain hydrology. While changing glacier areas are considered of highest importance for the understanding of future changes in runoff, glaciers are often
only poorly represented in hydrological models. Most importantly, the direct coupling between the simulated glacier mass balances and changing glacier areas needs feasible solutions. The use of a complex glacier model is often not possible due to data and computational limitations. The $\Delta h$-parameterization is a simple approach to consider the spatial variation of glacier thickness and area changes. Here, we describe a conceptual implementation of the $\Delta h$-parameterization into the semi-
distributed hydrological model HBV-light, which also allows for the representation of glacier advance phases, and comparison between the different versions of the implementation. The coupled glacio-hydrological simulation approach, which could also be implemented in many other semi-distributed hydrological models, is illustrated based on an example application.

## 1 Introduction

Glacier melt water is an important contribution to discharge in high-mountain catchments (Köplin et al., 2013; Miller et al., 2012) and can sustain summer streamflow in many large river basins (Hagg et al., 2007; Stahl et al., 2017). When modeling the hydrology of such catchments for longer periods (>10
years), the changing glacier area has to be considered, especially when climate change is causing glacier retreat. The simplest approach is to update the hydrological model with an externally simulated glacier extent, but this is unsatisfactory, as the mass balance simulated by the hydrological model might not agree with the updated glacier extent. The use of coupled glacio-hydrological models allows the glacier extent to be linked directly to the simulated glacier mass balance and is, thus, better suited for modeling

catchments with changing glacier areas (Huss et al., 2008; Stahl et al., 2008). However, modelers are faced with the question of which degree of complexity is needed to represent glaciers and glacier evolution in hydrological models. Several fully distributed, physically-based glacier models which consider mass balance, subglacial drainage, ice flow dynamics etc. have been developed over the past decades (e.g., Frans et al., 2016; Naz et al., 2014; Pattyn, 2002; Stroeven et al., 1989). While there are studies where such complex glacier models have been coupled with hydrological models (Frans et al., 2016; Naz et al., 2014), a simpler approach might be useful in many cases as the limited data availability would not allow the application of complex models, in particular their parameterization and validation. The use of such a complex model is also often too computationally expensive for use in a combined glacio-hydrological model where an entire catchment has to be considered. Many semi-distributed hydrological models use simplified representations of catchment hydrology using a limited number of conceptual buckets (reservoirs) and coupling such a model with a more complex glacier model would lead to a mismatch in degree of physical and spatial representation. Hence, for hydrological modeling studies there is a need for glacier models that use a similar degree of complexity and data demand as other components of the hydrological model but which are still able to represent the important glacier processes.

Recently an increasing number of hydrological models have incorporated glacier evolution models, using for example an equilibrium line altitude (ELA) shift (e.g., Linsbauer et al., 2012), volume–area scaling (e.g., Luo et al., 2013; Radić et al., 2008), —volume-area scaling and morphological image analysis (e.g., Stahl et al., 2008), other simple schemes without ice flow (e.g., Bongio et al., 2016), or more complex approaches focusing on glacier modeling (e.g., Immerzeel et al., 2012). One approach with limited glacier input data requirements, which is mass-conserving and well suited for hydrological modeling studies, is the $\Delta h$-parameterization, which describes the glacier thickness change at a certain elevation in response to an overall change in ice mass (Huss et al., 2010). Initially, Huss et al. (2008) introduced the $\Delta h$-parameterization as part of their Glacier Evolution Runoff Model (GERM), while a more detailed presentation of the approach, including the derivation of generalized empirical functions applicable to unmeasured glaciers, is given in Huss et al. (2010). Since then, the $\Delta h$-parameterization has been applied in global scale modeling by Huss and Hock (2015) as well as in numerous studies applying GERM to simulate individual glaciers or glacierized regions in the Swiss Alps (Farinotti et al., 2012; Finger et al., 2013; Gabbi et al., 2012; Huss et al., 2014; Huss and Fischer, 2016) and in Central Asia (Sorg et al., 2014). Several other glacio-hydrological models were coupled with glacier retreat simulations following the $\Delta h$-approach (Addor et al., 2014; Gabbi et al., 2014; Linsbauer et al., 2013; Ragettli et al., 2013; Salzmann et al., 2012; Vincent et al., 2014). However, details on its practical

implementation into the respective conceptual hydrological models have been provided by only few studies, for instance those by Li et al. (2015) and Duethmann et al. (2015).

As the $\Delta h$-parameterization is an empirical approximation to describe glacier retreat, it is subject to uncertainty and several limitations in terms of accurate glaciological modeling at the scale of individual glaciers (discussed in Huss et al., 2010; Linsbauer et al., 2013; Vincent et al., 2014). Nevertheless, for the purpose of transient hydrological modeling, particularly for regional studies covering large samples of glacierized catchments, the $\Delta h$-approach represents an efficient state-of-the-art alternative to more complex glacier evolution models (Huss et al., 2010; Li et al., 2015). Originally, Huss et al. (2010) derived the $\Delta h$-parameterization for periods dominated by negative mass balances and glacier retreat. The missing representation of glacier advance is related to uncertainties in regions with indications for a presence of recent glacier advance (Ragettli et al., 2013). Moreover, it represents a major drawback for long-term hydrological modeling covering past periods, for example the period with positive mass balance in the European Alps during the 1970s. A simplified scheme to incorporate short-term glacier change in case of advance as extension of the original $\Delta h$-approach is presented by Huss and Hock (2015).

Here, we describe a conceptual implementation of the $\Delta h$-parameterization into the semi-distributed hydrological model HBV-light (Seibert and Vis, 2012), which also allows the representation of glacier advance phases, and compare different versions of the implementation. This approach has recently been used to model a century of glacier runoff for 49 alpine catchments of the Rhine basin (Stahl et al., 2017). We present results from one of these catchments for illustration. This technical note aims at describing our implementation of the $\Delta h$-parameterization in such a way that researchers using other hydrological models also could follow the same approach. This follows the quest for reproducible science as recently emphasized for hydrological modeling (Hutton et al., 2016).

## 2 Materials and methods

### 2.1 New glacier routine

*HBV model and data requirements*

The HBV model is a semi-distributed conceptual precipitation–runoff model. It has continued to be developed in Scandinavia since the 1970s (Bergström, 1976; Lindström et al., 1997) and has become a standard tool which is widely used in different model variants, particularly for modeling snow-dominated catchments. Required input data are daily temperature, precipitation and potential evaporation time series. Additionally for the new glacier routine information on the initial glacier areas

and ice thickness values, both as a function of elevation, are required. For the estimation of these initial conditions glaciologists have developed a number of approaches as recently reviewed by Farinotti et al. (2017). One possible method is described in Appendix A1.

In the HBV model the hydrological processes within a catchment are modeled by four different routines, a snow/glacier routine, a soil moisture routine, a response routine and, finally, a streamflow routing routine. Here, we describe the recent integration of a glacier evolution approach into the HBV-light software, a user-friendly and freely available version of HBV (Seibert and Vis, 2012).

*Snow and ice accumulation, melt and runoff*

The glacier area within a catchment is conceptually simulated by two reservoirs representing glacier ice and the liquid water contained within the glacier. There can be a snowpack on top of the glacier, which also consists of a solid (snow) and a liquid (water content) reservoir. The snow and glacier routine calculations are performed at each simulation time step for each elevation zone, for which elevation intervals of 100 to 200 m are typically used. The elevation zones can be further subdivided according to three aspect classes (N, S, and W/E). Depending on the temperature in relation to the threshold temperature, precipitation falls either as snow or rain. In case of rain, the precipitation is added to the water content of the snow if a snow layer is present or to the water content of the glacier otherwise. If the temperature is above the threshold temperature, melt takes place in the snowpack based on a degree-day factor, and the melted snow is added to the water content of the snowpack. In the case that the water content exceeds the snow water holding capacity, the amount exceeding the snow water holding capacity flows out and is added to the liquid water reservoir of the glacier. If the temperature is below the threshold temperature, part of the water content in the snow layer refreezes. The use of aspect classes allows considering the faster respective slower snow melt in certain parts of the catchment by applying an additional aspect factor to the degree-day equation (Hagg et al., 2007; Hottelet et al., 1993), which taken together leads to a prolonged but less intense melt period at the catchment scale compared to the situation when not using different aspect classes.

For ice melt of the glacier a degree-day method is used as well, but ice melt is only simulated at times when there is no snow layer on the glacier. For temperatures above the threshold temperature, glacier melt is calculated using the degree-day factor multiplied with a glacier correction factor, which represents the different albedo of ice compared to snow and typically takes values of about 1 to 2 (Hock, 2003). The ice melt is added to the liquid component of the glacier, from which the outflow is computed individually for each elevation zone as suggested by Stahl et. al. (2008) extending earlier concepts by Moore (1993), to account for the enlargement of glacial conduits over the melt season.

$$Q(t) = S(t)(K_{min} + K_{range} \cdot e^{-A_G \cdot S_{WE}(t)}) \tag{1}$$

$Q$ is the outflow, $S$ the liquid water content of the glacier, , the parameters $K_{min}$ and $K_{range}$ are the minimum outflow coefficient and maximum range of outflow coefficient values, and $A_G$ a calibration parameter controlling the outflow response dependent on $S_{WE}$, which is the water equivalent of the snowpack on the glacier. To represent the transition from snow to firn in a simple way, at the end of each time step a certain fraction of the snow on top of the glacier is converted into firn and equally distributed over the whole glacier area. Typical values for this model parameter are 0.001–0.003 which implies that the conversion of snow to firn on average takes about 1 to 3 years (Luo et al., 2013). The further transition from firn to ice takes place over much longer time periods from 10 to over 100 years. For the glacier modeling presented here, however, firn is considered as a part of the accumulated glacier mass.

Snow redistribution by wind and avalanches can be important to consider in modeling alpine catchments (Freudiger et al., 2017a). Therefore, in our modeling approach optionally snow redistribution can be applied at the end of each time step to avoid unrealistic multi-year snow accumulation, the so-called "snow towers". As snow redistribution was not the focus of this study, we used a simple approach. During the snow redistribution, the snow (i.e. snowpack and snow water content) of all non-glacier areas above a certain user-specified elevation, $H_{redist}$ , and after reaching a certain user-specified $S_{WE}$ threshold, is redistributed evenly over the non-glacier and glacier areas within a user-specified elevation range below $H_{redist}$, as well as the glacier areas above $H_{redist}$.

*Glacier mass and area changes*

The technical details of the implementation of the new Glacier Area Change Routine (GACR) into HBV-light are outlined in a flowchart (Figure 1). To translate glacier mass changes into area changes , a single-valued relation between glacier mass and glacier area needs to be established. This relationship, which is technically represented in the model by a lookup table, can be derived from any glaciological model. Here we suggest that the relationship (and lookup table) is computed based on an initial variation of glacier thickness values with elevation (termed initial glacier profile in the following) and the $\Delta h$-parameterization method described in Huss et al. (2010), scaling the relative elevations to those of the study catchment (Figure 2). For these calculations each elevation zone (of typically 100–200 m) in the model application is further subdivided into elevation bands (of typically 10 m) to ensure smooth changes. The use of a lookup table enables the representation of periods of glacier advance (though not further than the initial glacier extent).

The basic idea is that the total glacier volume, $M$, is defined by integration of the initial glacier profile (Eq. 2):

$$M = \sum_{i=1}^{i=N} a_i \cdot h_i \qquad (2)$$

$M$ is the total glacier mass in mm water equivalent relative to the entire catchment area, and for each

elevation band $i$, the area $a_i$ (expressed as proportion of the catchment area) and water equivalent $h_i$ in mm. To generate the lookup table the glacier then is melted in steps of $\Delta M$. For each of these steps the $\Delta h$-parameterization method of Huss et al. (2010) is applied. For each elevation band the normalized elevation, $E_{i,norm}$, is computed from the absolute elevation $E_i$ of the corresponding elevation band $i$, as well as the maximum and minimum elevations of the glacier, $E_{max}$ and $E_{min}$ (Eq.3).

$$E_{i,norm} = (E_{max} - E_i)/(E_{max} - E_{min}) \qquad (3)$$

The normalized water equivalent change is then computed for each of the normalized elevations using the following function (Huss et al., 2010):

$$\Delta h_i = (E_{i,norm} + \alpha)^\beta + (E_{i,norm} + \alpha)^\beta + \delta \qquad (4)$$

where $\Delta h_i$ is the normalized (dimensionless) ice thickness change of elevation band $i$ and $\alpha$, $\beta$ and $\delta$ are

empirical coefficients. Based on the initial total glacier area (in km²) that needs to be specified in addition to the initial glacier thickness profile, one of the three empirical parameterizations applicable for unmeasured glaciers from Huss et. al. (2010) is used (Figures 1 and 2a).

In the next step a scaling factor $f_s$ (mm), which scales the dimensionless $\Delta h$, is computed based on the glacier volume change $\Delta M$, and on the area and normalized water equivalent change for each of the

elevation bands:

$$f_s = \Delta M /(\sum_{i=1}^{i=N} a_i \cdot \Delta h_i) \qquad (5)$$

The new water equivalent $h_{i,k+1}$ is then computed for each elevation band starting from the user-specified initial glacier thickness profile for $k=0$ as

$$h_{i,k+1} = h_{i,k} + f_S \Delta h_i \qquad (6)$$

where $h_{i,k}$ is the water equivalent of elevation band $i$ after reducing the glacier mass $k$ times by $\Delta M$. Exemplary results of this step-wise melt process based on the $\Delta h$-parametrization are visualized in Figure 2b.

Once the new water equivalent values have been computed for each elevation band, the glacier area is updated for each elevation zone. The relative glacier area for a certain elevation zone is defined as the cumulative area of the glacier covered elevation bands within that elevation zone, divided by the total area of the elevation zone. Thus the model described so far is essentially a 2-D representation of glacier retreat. However, glaciers have an uneven distribution of ice at a particular elevation with a thinner ice layer along the edges. In order to take the area reduction that results from this uneven distribution into account, a simplified representation of the 3-D glacier geometry is used to scale the area within a certain elevation band (Eq. 7) following the relation between glacier width and glacier thickness given in Bahr et al. (1997) as also applied by Huss and Hock (2015):

$$a_{i,scaled} = a_i \cdot \sqrt{h_i / h_{i,initial}} \tag{7}$$

The reduction in glacier area over elevation resulting from the application of the $\Delta h$ parameterization following Eq. 2–6 in combination with the glacier width scaling (Eq. 7) is illustrated in Figure 2c. The resulting relationship between glacier area and glacier mass is stored in the lookup table at steps of 1% of the initial glacier mass. This means that the lookup table consists of glacier areas per elevation zone for 101 different glacier mass situations ranging from the initial glacier mass to zero (Fig.1).

During the actual simulation in HBV-light, the glacier extent is updated at the beginning of each hydrological year  (1. October). The total water equivalent of the glacier is computed. Based on the percentage of glacier water equivalent in comparison to the total glacier water equivalent in the initial glacier profile definition, the corresponding record is extracted from the glacier lookup table and the corresponding glacier areas are applied to the different elevation zones. In the case that the glacier water equivalent exceeds its maximum, the areas corresponding to 100% are applied (i.e. the glacier can never grow larger than defined by the user in the glacier profile definition). Optionally simulations can start, however, with a reduced glacier size, by specifying the initial glacier fraction in the glacier profile file (as fraction of water equivalent). The initial glacier profile definition should thus contain the maximum extent of the glacier during the full simulation period. For each glacierized part of an elevation zone in HBV-light, the corresponding non-glacierized part is used to exchange the area for which the state changed from glacier to non-glacier and vice versa. In order to ensure the water balance is correct, bookkeeping is done between the corresponding glacierized and non-glacierized zones. Soil moisture

and snow, for example, are moved between the corresponding zones as far as it corresponds to the area exchanged.

## 2.2 Sensitivity test of different model variants

To illustrate the new Glacier Area Change Routine (GACR) and its different components on the simulation results, we applied the HBV-light model for one example catchment, the Alpbach catchment in the Swiss Alps. This catchment is one of the glacierized headwater catchments in the Rhine River basin, located in Central Switzerland. The catchment area is about 21 km² and elevations range from 1022 m up to 3192 m a.s.l. with a mean elevation of 2194 m. The catchment consists of two main valleys with the glacier Glatt Firn extending into both of them. According to the glacier inventory for the year 2010 the glacierized area was estimated 4.03 km² ( Fischer et al., 2014), whereas the estimate was 4.54 km² for 2003 (Paul et al., 2011), corresponding to a catchment glacier coverage of about 20%. The initial glacier profile for 1900 was estimated as described in Appendix A1.

To demonstrate the effect of the different parts of the GACR, four different versions of the GACR were used, where for three versions certain components of the new glacier routine were disabled:

**1) Stationary glacier area (No GACR)**

Only the static part of the glacier routine is used, i.e. the complete dynamic part of the glacier routine is disabled; the glacier area is not adjusted but stays exactly as defined by the user in the model setup during the whole simulation.

**2) Full new GACR (GACR)**

The full version of the model as described in section 2.1, with the static and dynamic part of the glacier routine included.

**3) GACR without glacier width scaling (GACR-w)**

The application of glacier width scaling (Eq. 7) by elevation band is disabled. In practice, this corresponds to a 2-D representation of glacier area change. A change in glacier area is realized only when the mean glacier water equivalent of an elevation band (Eq. 6) reaches zero. As a result, the area change will only occur at the glacier terminus.

**4) GACR without glacier advance – glacier retreat only (GACR-a)**

The original method described by Huss et. al. (2010) considers only the parameterization of glacier retreat and not glacier advance. In the new GACR, glacier advance up to the initial state is enabled

by means of the lookup table generation. To demonstrate the effect of neglecting temporary glacier advance, we used a version that applies only glacier retreat. In periods with a positive annual glacier mass balance the glacier area is kept constant.

For each of these four versions, we calibrated the model 10 times, using a genetic algorithm (Seibert, 2000) with 3500 model runs per calibration trial. The 10 independent calibration trials allowed parameter uncertainty effects to be considered by taking several optimized parameter sets into account. The simulation period was January 1$^{st}$, 1901 to December 31$^{st}$, 2006, and was preceded by a three year spin-up period. As an objective function, the average of the Nash-Sutcliffe efficiency for daily

discharge, the relative volume error of the total discharge, root mean squared error of the snow cover simulations and absolute mean relative error of the glacier volume estimates were used. The estimates of observed glacier volume were based on different glacier cover datasets for three particular years during the simulation period as described below.

The simulation period (1901–2006) is a period in which glaciers of the European Alps retreated

considerably, yet this period also covers diverse climate conditions including a period between the 1960s and the 1980s that was characterized by rather balanced conditions or temporarily by glacier advance. For the setup and the calibration of the model in terms of glacier conditions, several observation based datasets from diverse sources were used: the glacier area for the state around the years of 1901 (start of simulation period) and 1940 was based on digitized historical topographic maps,

in Switzerland known as "Siegfriedkarte". For both years, 1901 and 1940, the area of the Alpbach catchment is reconstructed from two adjacent map sheets. To describe the glacier area around the simulation start in 1901, maps from the years 1894 and 1899 were used, and to describe the glacier area around 1940, maps from the years 1933 and 1942 were used. Glacier areas for the years 1973, 2003, and 2010 were extracted from the glacier inventories by Müller et al. (1976), Paul et al. (2011) and

Fischer et al. (2014), respectively. For the years 1973 and 2010 gridded datasets of estimated glacier thickness based on the method presented in Huss and Farinotti (2012) were also used (unpublished data provided by Matthias Huss). In addition, discharge observations (station Erstfeld, Bodenberg, period 1960–2006) from the Swiss Federal Agency of the Environment (FOEN) and a gridded snow water equivalent (SWE) climatology product from the Institute of Snow and Avalanche Research (WSL-SLF,

covering Nov–May for the period 1972–2006) were used to calibrate the model. More details on the underlying data sources and the applied multi-criteria calibration can be found in Stahl et al. (2017).

To setup the HBV-light model for the Alpbach catchment, the spatial modeling units were discretized as follows: Firstly, the glacierized and non-glacierized catchment area fractions for the state at simulation

start 1901 were distinguished. Therefor all areas within the Alpbach catchment that were glacier covered according to the underlying map or glacier inventory for a specific year were summed up as one model glacier. Both the non-glacierized and the glacierized model areas were further divided into area fractions per elevation zones, and then further differentiated within each elevation zone into area fractions for three aspect classes.

For the application of the $\Delta h$-parametrization, in addition to the main model setup the initial glacier profile needs to be defined by the user (Figure 1). As no data on glacier thickness for the state at 1901 (start of simulation) were available, an initial glacier profile had to be reconstructed; details for the method, which was chosen in this application, are described in the appendix. The reconstructed glacier profile used for model initialization is shown in Figure 2b (black line for M = 100%).

**3 Results**

Figure 3a shows the observed glacier profile for the initial state at simulation start in the year 1901 as well as for the three different years for which data was available from which the glacier profile could be derived. The observed decrease of glacier area occurred at all elevations. Figure 3b shows the glacier profile for the simulation with the full new GACR model version. With this version, glacier retreat also occured at all elevations. This is due to the combination of the $\Delta h$-approach and the implemented width scaling. In order to compare the simulated and observed glacier profiles, Figure 3c shows the differences between simulated and observed glacier area for the different elevation zones. The $\Delta h$-approach by definition results in zero change in glacier thickness at the very top of the glacier. The lower the position on the glacier, the larger the change in thickness can be. This pattern can be seen in Figure 3b, where there is, contrary to the observed data of Figure 3a, hardly any change in glacier area in the higher elevation zones. As a result, the difference between simulated and observed areas in Figure 3c is positive for the higher elevation zones (for the years 1973 and 2003). This is compensated for by a negative difference between simulated and observed glacier areas for the lower elevation zones. Overall, the new GACR is able to depict the major pattern of long-term glacier area change over the elevation zones in the example catchment.

The simulations using the full GACR also correspond well with the observations in terms of total catchment glacier area (Figure 4a). In addition, Figure 4 illustrates the differences in the four different model versions to simulate the changes in total catchment glacier area (Figure 4a) and the resulting effects on the change in glacier water equivalent and cumulative glacier runoff (Figure 4b and c), which are relevant within the scope of hydrological modeling. Among all model versions the new full GACR is best in representing the pattern of change in total glacier area based on the comparison with available

observation based data (Figure 4a). Whereas there is a considerable mismatch of the simulated and observed glacier area around the year 1940, for the later years the simulated and observed glacier areas are in good agreement. The model version that does not incorporate glacier advance is just as effective in meeting the final state of the glacier area in the year 2003 as the full version. In terms of glacier area the results of both versions, GACR-a and GACR, are only different during phases dominated by positive glacier mass balance. As soon as the annual glacier water equivalent (glacier volume) decreases to its previous minimum again, the reduction in glacier area continues. For glaciers with a net negative mass balance over time, differences can therefore be rather small. If there are more and longer periods of glacier advance, differences might become more apparent. However, in case of overall net positive glacier mass balance, the fact that the maximum glacier extent cannot exceed what is specified in the glacier profile file becomes an obvious limitation of the new GACR routine. For the version GACR-w the glacier stays at its maximum size a bit longer than for the full new GACR and the version GACR-a, since elevation bands need to be melted completely before the glacier area starts to reduce. In contrast, in the full new GACR and GACR-a simulations width scaling is applied as soon as the glacier mass balance becomes slightly negative, and therefore a reduction in glacier area can be observed immediately. For simulations with only the static glacier routine (No-GACR) the glacier area stays constant (horizontal grey line in Figure 4a).

The constant area with the No GACR version allows for (much) higher melt rates in comparison to the other model versions once the glacier has partly melted, since a larger area is contributing to the overall melt than in the version including a GACR. This can also be clearly observed in Figure 4b, where the model version with a stationary glacier area shows much stronger glacier water equivalent changes. As a result the cumulative glacier runoff (Fig. 4c) is highest for simulations with No GACR, especially during the second half of the simulation period when the difference in glacier area in comparison to the other versions is more notable. Generally, the larger the glacier area, the more runoff is generated by the glacier. The stationary glacier area model (No GACR) results in the potentially largest amount of glacier runoff, followed by the simulations without width scaling (GACR-w), for which the 10 different model calibrations resulted in the largest spread. The difference between the versions GACR and the GACR-a is minor, with the latter likely resulting in an underestimation of generated glacier runoff, due to the smaller area during phases of glacier growth.

**4 Discussion and conclusion**

The glaciological part of the coupled model as described above is a simple representation of glacier processes, but it allows glacier geometry changes to be considered at a level of complexity which is similar to the hydrological model. In most current hydrological models no representation of changing

glacier areasis realized, which basically implies an infinitely thick glacier. The approach described here, which allows for area changes as a result of simulated mass changes, is certainly a more realistic representation and the changing area clearly affects variables such as the simulated runoff. Some previous studies used a simple volume–area scaling (e.g. Luo et al., 2013). This approach does not consider any catchment-specific information, whereas the $\Delta h$-parameterization allows considering elevation distributions and the ice thickness profile. In volume–area scaling any volume change directly translates to area changes, although this has not always to be the case. The $\Delta h$-parameterization also allows attributing the glacier area changes to the different elevation zones, which would not be directly possible with a simple volume–area scaling, which does not allow assigning the region of glacier shrinkage (see also the discussion by Stahl et al. (2008)). As discussed by Huss et al. (2010) the $\Delta h$-approach is a simple but still physically based approach to consider changing glaciers as a result of the simulated mass balance change.

A major simplification of the approach presented here is that only one glacier is considered in each subcatchment, which means that if there are several glaciers these are simulated as one virtually aggregated glacier. Principally this could be solved by using as many subcatchments as there are glaciers. However, this would not solve the issue of a glacier which splits up into several glaciers at some point during the simulation. The representation of all glaciers in a catchment as one virtually aggregated glacier might, thus, be a suitable representation. The $\Delta h$-parameterization approach of Huss et al. (2010) and the use of their empirical functions were found suitable. This reduces the need for new calibration parameters. While the $\Delta h$-parametrization could also be based on data for specific glacier(s) (as done, for instance, by Duethmann et al., 2015), which would better represent local conditions, in reality such data is rarely available. A re-evaluation of the empirical $\Delta h$-parameterizations, which included glaciers from different parts of the world, resulted mainly in satisfying results (Huss and Hock, 2015).

Several adoptions were needed for the implementation in a semi-distributed model. Most importantly the use of a lookup table to represent the mass–area relationship allows for advancing glaciers. Furthermore, the geometric width scaling for individual elevation bands allows for the representation of a decreasing glacier area with decreasing thickness in an elevation band. The example simulations shown in this technical note illustrate the effect of these modifications, which maintain the conceptual model approach. Allowing for advancing glaciers and changing areas due to glacier thinning makes a difference in the simulations (Fig.4). Both these aspects are also important as they enable a comparison between simulated and observed glacier area (see Fig. 3a and Fig.2). This is crucial for model calibration and validation as glacier areas and glacier lengths are much more frequently available than other glacier observations. The simulations demonstrate that the new glacier evolution routine is, in

general, capable of simulating reasonable area changes. However, given the limited data this should not be taken as a proof that the model is correct, even if the simulations appear glacio-hydrologically reasonable. The validation of any glacier model or routine against observations is challenging due to limited suitable data sets and beyond the scope of this technical note.

Besides its simplicity, the presented GACR implementation also has other limitations. One challenge is to obtain initial thickness distributions along the glacier. While this estimation of initial glacier conditions certainly adds uncertainties, information on initial ice thicknesses is needed for any approach that aims for simulating changing glacier areas. In the approach presented here, glacier advance is only possible up to the initial state. In most cases this is not a major limitation as long as suitable information

on early glacier extents is available as most climate data and scenarios lead to retreating glaciers. If needed, a larger initial glacier extent (with some thickness profile) can be provided to establish the mass–area relation to create the lookup table. In this case the actual simulations would start at a certain fraction of this hypothetical maximum situation.

The $\Delta h$-parameterization represents an approach, which allows changing glacier areas to be considered

in an approximate but realistic way. The conceptually stringent implementation presented in this technical note could in principle also be used by other semi-distributed hydrological models. In many hydrological model applications of partially glacierized catchments that do not specifically target the contributions of glaciers to runoff, glacier areas are not directly updated. Studies with a coupled glacio-hydrological approach often describe little details of the glacier routine, especially when it comes to the

question on whether simulated mass balance changes are translated into glacier area changes and, if so, how this is done. In a recent review on hydrological modeling of glacierized catchments in central Asia (Chen et al., 2016), for instance, this issue is not discussed at all. The main advantage of the coupled glacio-hydrological approach as described in this technical note is that glacier mass and area changes are consistent with the hydrological model. This also allows the model to be used to simulate future

scenarios. While the GACR described in this technical note is a rather simple representation of glacier processes, it enables this important representation of changing glacier areas in high mountain catchments.

### *Acknowledgments*

We thank Daphné Freudiger and Damaris De for their contributions including the digitization of glacier

outlines from historical maps. Matthias Huss provided details on the original $\Delta h$-method and on the estimation of the glacier profiles as well as ice thickness data. The model code extension was made possible with funding by the University of Zürich. The method developments were made within the ASG-Rhein project (The snow and glacier melt components of the streamflow of the River Rhine and

its tributaries considering the influence of climate change) funded by the International Commission for the Hydrology of the Rhine Basin (CHR).

**Data availability**

Meteorological data input used was the HYRAS interpolation product made available by the German Weather Service DWD and the Bundesanstalt für Gewässerkunde BfG and a reconstruction HYRAS-REC by Stahl et al. (2017). Climate station data was provided by MeteoSwiss. Model calibration used hydrometric data by the Swiss Federal Office for the Environment FOEN, snow data of the "SLF-Schneekartenserie Winter 1972-2012" by the SLF (WSL Institute for Snow and Avalanche Research) and glacier data provided by Matthias Huss and glacier areas from the Siegfriedkarte by Swisstopo.

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

**Figure captions**

Figure 1. Flowchart describing the update of the glacier geometry depending on glacier mass balance changes in HBV-light. Additional information is given in the following notes (numbers refer to corresponding numbers in the flowchart):

(*1): Elevation bands and corresponding water equivalent, with elevation bands at a finer resolution than the elevations zones. While the areal distribution of a static glacier is specified in HBV-light by means of elevation and aspect zones, for establishing the relationship between glacier mass and glacier area a glacier profile, which defines the initial thickness (in mm water equivalent ) and areal distribution of the glacier at a finer resolution, is needed as model input data. Note that the resolution of the glacier

routine simulations largely depends on the number of elevation bands per elevation zone, i.e. all glacier area within each band is either covered by a glacier or not, and the percentage of glacierized area within a certain elevation zone is based on the state of the individual elevation bands within that elevation zone. Elevation zones typically have resolutions of 100 to 200 m, whereas for the elevation bands a resolution of 10 m is commonly used.

(*2): Huss et al. (2010), which of the three parameterizations is used depends on the glacier area (see Eq. 4)

   (*3): Reduce for each elevation band the glacier water equivalent according to the empirical functions from Huss et al. (2010) (Eq. 4) to compute the glacier geometry for the reduced mass (see Eq. 6). If the computed thickness change is larger than the remaining glacier thickness (most likely to occur at the

glacier tongue, see the area that is marked in red in the figure), the glacier thickness is reduced to zero resulting in a glacier-free elevation band, and the portion of the glacier thickness change that would have resulted in a negative glacier thickness is included in the next iteration step (i.e. the next 1% melt).

   (*4): The Δh-approach distributes the change in glacier mass over the different elevation zones, though results in glacier free areas mainly at the lowest elevations. The width scaling within each elevation

band relates a decrease in glacier thickness to a reduction of the glacier area within the respective elevation band. In other words, this approach allows for glacier area shrinkage also at higher elevations which mimics the typical spatial effect of the downwasting of glaciers.

   (*5): Define elevation zones and compute for each elevation zone the fractions of glacier and non-glacier area (relative to the catchment area)

(*6): Sum for each elevation zone the total (width scaled) areas for all respective elevation bands which are covered by glaciers (i.e., glacier water equivalent ≥0)

(*7): M (in % of initial M) in first column followed by one column for each elevation zone with the areal glacier-cover area (in % of catchment area)

(*8): Run once before the actual simulation of time series starts (automatically within the HBV-light software)

Figure 2: $\Delta h$-parameterization and its implementation in HBV-light: a) empirical $\Delta h$-parameterization functions for three glacier size classes from Huss et al. (2010), b/c) pre-simulation application of the $\Delta h$-parameterization for a medium glacier size to the example glacier profile data of the Alpbach catchment by melt in steps of $\Delta M = 1\%M$ to generate the lookup table. b) absolute glacier volume per elevation band, c) relative glacier area per elevation band as relative fraction of the initial glacier area $a_{initial}$ of the elevation interval.

For each elevation interval, the resulting glacier water equivalent/glacier volume (b) and glacier area (c) are shown as grey lines, for visibility, only results of steps of $\Delta M = 5\%$ are shown here. The initial profile (100% $M$) and profiles for a glacier volume reduction by 20, 40, 60, and 80% are highlighted by colored labeled lines.

Figure 3: Observed and simulated glacier areas per elevation zone for years when observations are available: a) Glacier areas for the different elevations derived from maps or remote sensing, b) Glacier areas for the different elevations as simulated with the full new GACR model version, c) Differences between observed and simulated glacier areas.

Figure 4: Comparison of the simulations by the different versions of the glacier routine: a) Total glacier area, b) Change in glacier storage, c) Cumulative glacier runoff. The range of simulation results represents the results from 10 model (equally suitable) parameterizations for each of the different versions of the glacier area change routine (GACR) and for the version without glacier area change routine.

**Figures**

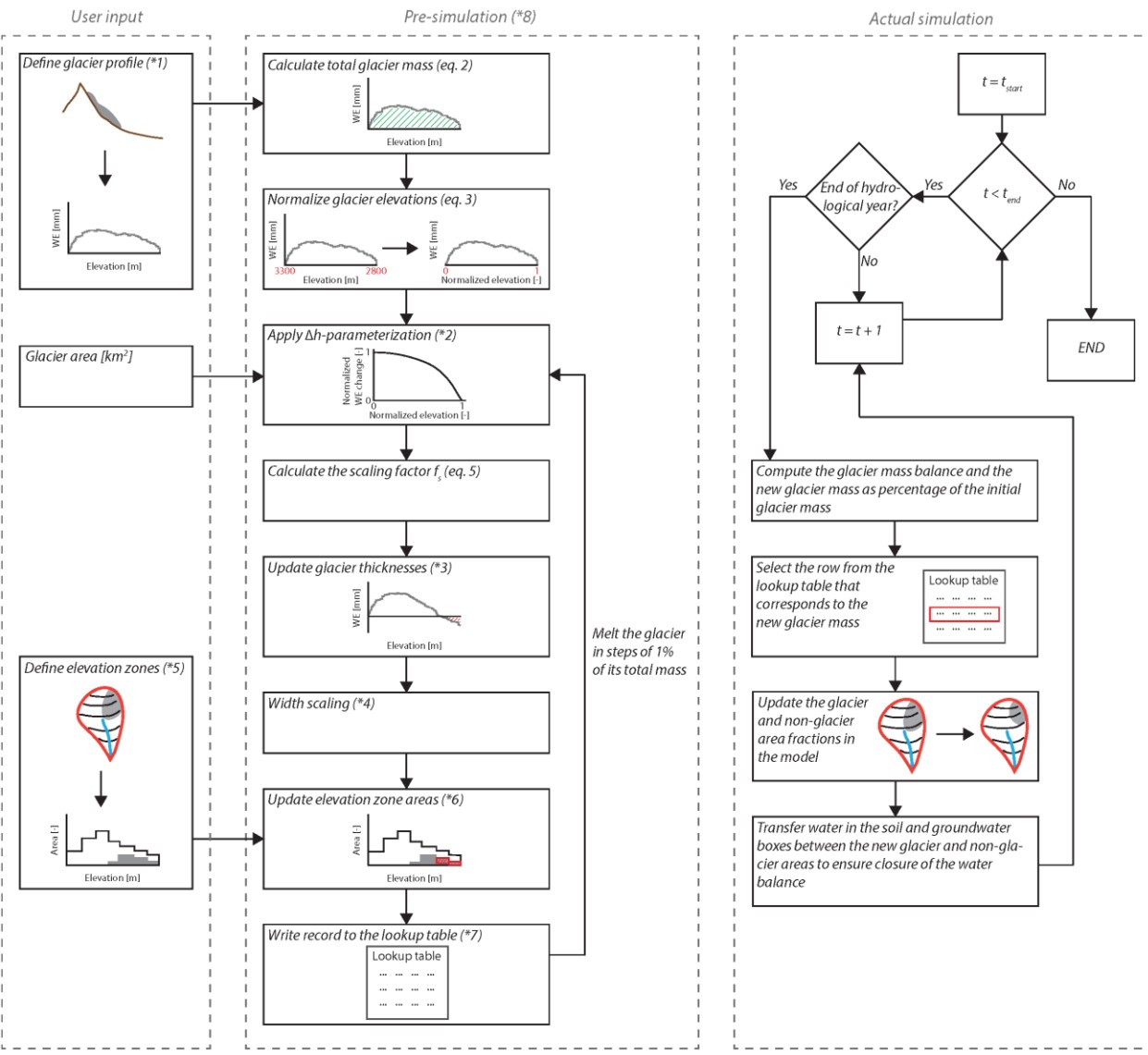

Figure 1.

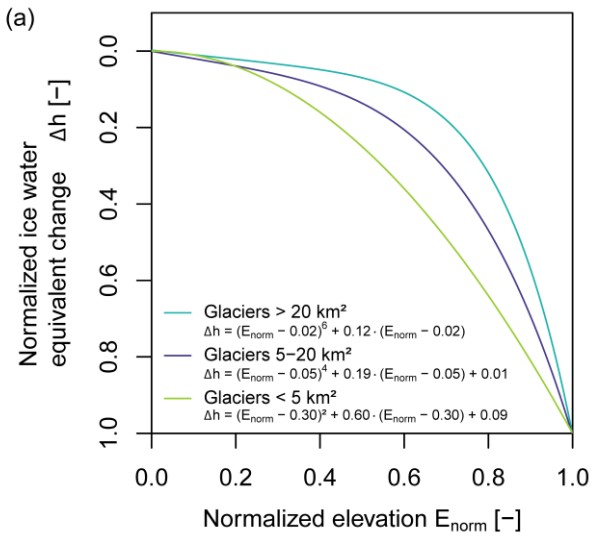

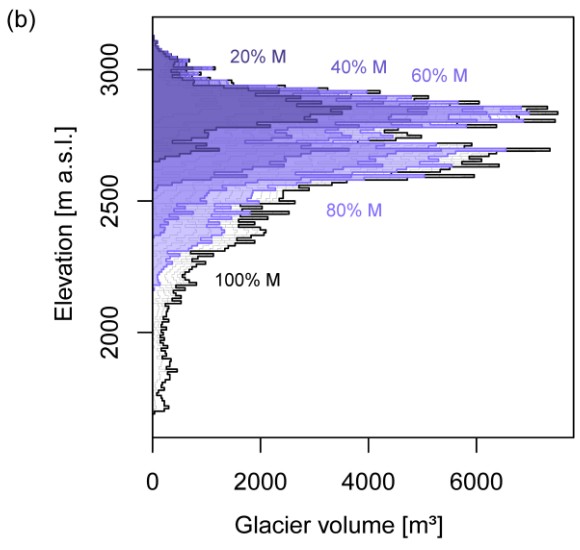

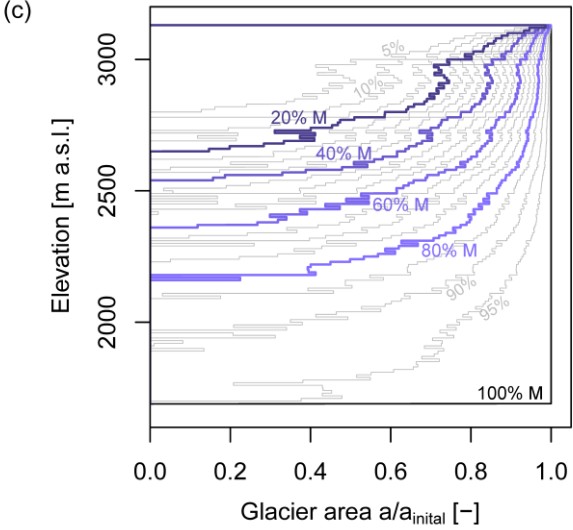

Figure 2

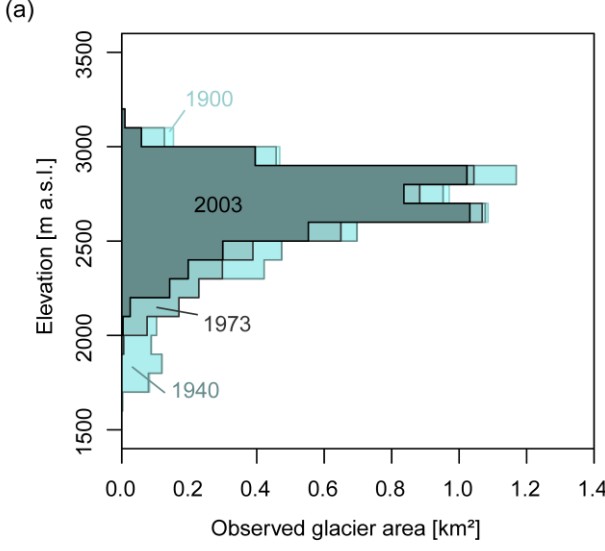

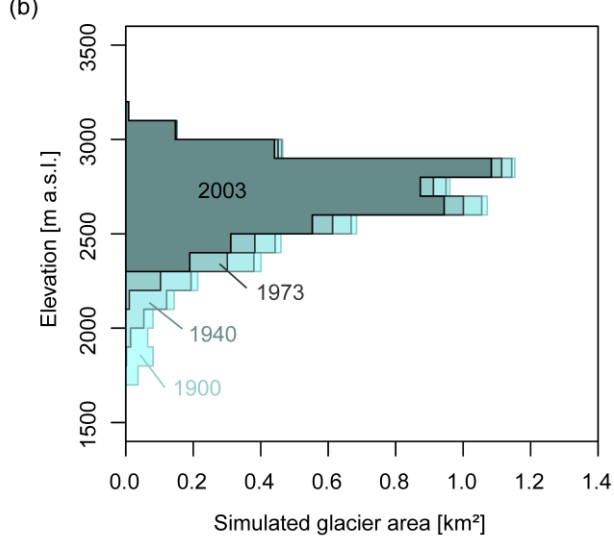

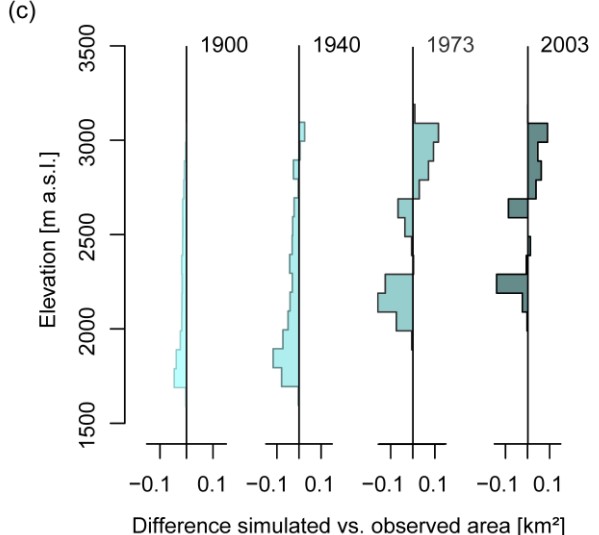

Figure 3

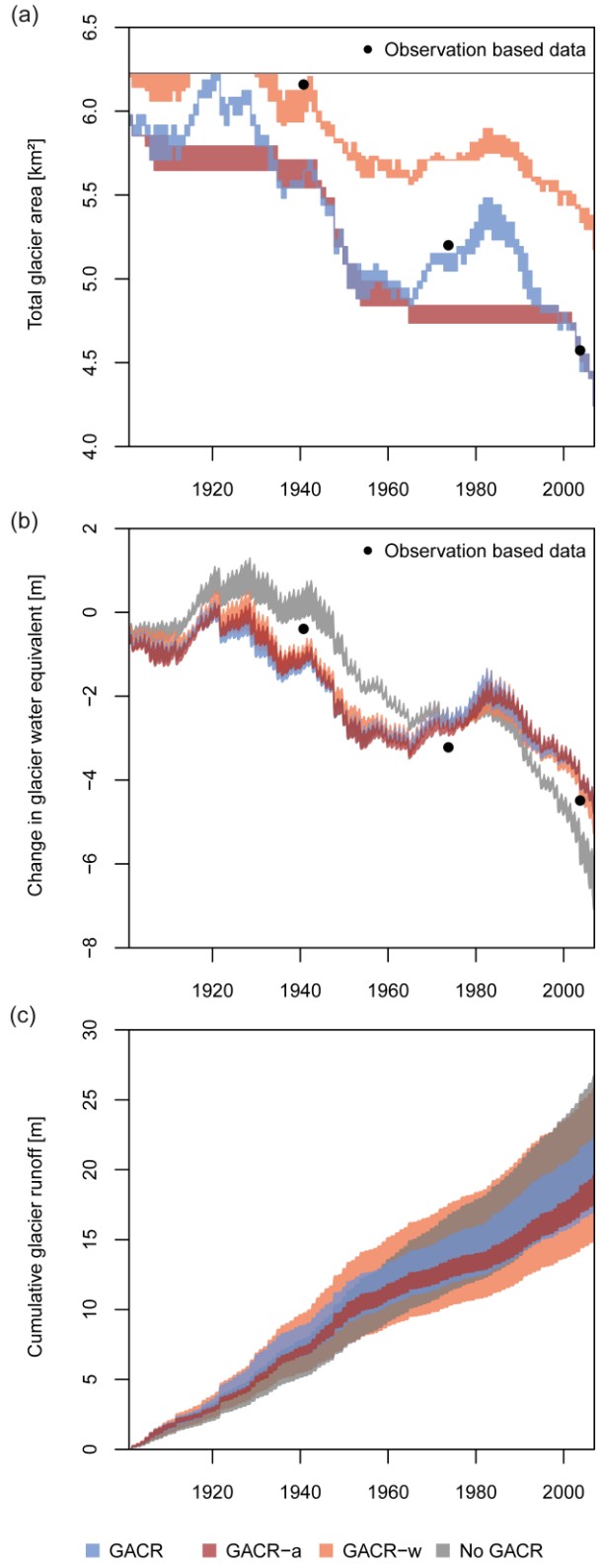

Figure 4

## Appendix A1: Reconstruction of initial glacier geometry

A challenging requirement for the application of the new HBV-light GACR, as for any modeling of temporally changing glacier geometry, is the definition of the initial state of the glacier in terms of total volume and ice thickness distribution, briefly termed initial glacier profile in the following. Approaches to tackle this, as recently reviewed by Farinotti et al. (2017), strongly depend on the available glacier survey data. For the case of the Alpbach catchment a reconstruction of the initial glacier profile for the state around the model simulation start year 1901 was required. Table A1 summarizes all available primary glacier data sets with reference to their origin as well as derived data used for the reconstruction of the initial glacier profile.

The glacier profile finally needed in the HBV-light setup consists of glacier area and thickness per elevation band. Whereas such data is available for the more recent years 1973 and 2010, for 1901 glacier area was the only information. Generally, the approach to estimate the initial ice thickness distribution was based on two physically-based glacier scaling relationships taken from Bahr et al. (1997): i) the widely applied general volume–area scaling relation (Eq. A1) and ii) a proportionality of glacier width and the square root of glacier thickness. The latter relationship assumes a parabolic cross section as characteristic for valley glaciers and was also used for the implementation of the new GACR (Eq. 7 in the main text). In detail, for the reconstruction of the initial ice thickness distribution the total glacier volume around 1901 was estimated based on:

$$V = c \cdot A^{\gamma} \tag{A1}$$

where $V$ is the total glacier volume (m³), $A$ is total glacier area (m²), $c$ is a glacier specific scaling parameter (m), and $\gamma$ is the scaling exponent (-), which was fixed to its theoretically defined value (Bahr et al., 2015) of $\gamma = 1.375$. The multiplicative scaling parameter $c$ for both glacier volume–area pairs (Table A1), for years 1973 and 2010, was obtained. The average of both values of the multiplicative scaling parameter $c$ was then used to estimate the total glacier volume for the simulation start in 1901 using the known glacier area (Table A1). To reconstruct the glacier thickness distributions over the elevation bands (10 m resolution in the example of the Alpbach), the proportionality of glacier width and the square root of glacier thickness was then applied to the elevation bands. The glacier width of an elevation interval can be used to approximate the glacier area of the elevation interval $i$ with:

$$A_i = p_i \cdot \sqrt{H_i} \tag{A2}$$

where $A_i$ is the glacier area (m²), $H_i$ is glacier thickness (m), and $p_i$ is a scaling parameter (m$^{1.5}$). Based on Eq. A2 the glacier and elevation band specific glacier width scaling parameters $p_i$ were determined

for the available observation-based 'glacier profiles' ($A_i$ and $H_i$ for all elevation bands $i$) for the years 1973 and 2010. With the values for the year 1973 a power law function was fitted to estimate the glacier width scaling parameter $p_i$ as a function of $A_i$. The obtained function was then used to estimate the initial glacier thickness $H_{i, 1901}$ for all elevation bands based on $A_{i, 1901}$. Finally the resulting estimated glacier

thickness values were corrected by a factor to enforce that the resulting total glacier volume ($\sum A_{i, 1901} \cdot H_{i, 1901}$) equals the total glacier volume estimate derived for the year 1901 from Eq. A1 above (Figure A1). With that, glacier area $A_{i, 1901}$ taken from the historical map (Table A1) and estimated glacier thickness $H_{i, 1901}$, the tabular glacier profile to initialize the HBV-light model simulations was generated. For the use in HBV-light, the glacier area $A_{i,}$ needs to be expressed as fraction of total catchment area ($a_i$

, -), and glacier thickness $H_{i,}$ is converted to water equivalents ($h_i$, mm) by applying an ice density of 900 kg m$^{-3}$, and the elevation bands $i$ (10 m intervals) are assigned to the corresponding elevation zones (100 m intervals) of the HBV-light catchment discretization.

One should note that the presented procedure to estimate the initial glacier geometry is subject to several uncertainties and limitations. These are, for instance, related to the uncertainties of the

underlying data sources, the treatment of several glacier parts or branches as one aggregated glacier, the application of the average of the glacier scaling parameter $c$ for the years 1973 and 2010 to estimate the glacier volume in 1901, the negligence of changes in surface elevation, or the fact that results obtained from glacier scaling applications on individual glaciers should always be regarded as an order of magnitude estimate only. However, as such, a way to get a rough estimate for glacier initialization, it

may still considered a feasible and reasonable approach for many hydrological model applications in glacierized catchments and in particular large catchment sample modeling studies facing a lack of detailed glacier survey data.

**Table A1: Glacier data sets with reference and derived data for the reference years 1900, 1973, and 2010 used for the reconstruction of initial glacier geometry for the Alpbach catchment.**

| Reference Year (ca.) | Reference | Original data | Derived data |
|---|---|---|---|
| 1901 | (Freudiger et al., 2017b) | Glacier outlines[1] | Total glacier area |
| | | | Glacier area per elevation band[3] |
| 1973 & 2010 | Matthias Huss (unpublished data) | Gridded ice thickness data[2] | Total glacier area |
| | | | Glacier area per elevation band[3] |
| | | | Total glacier volume |
| | | | Mean thickness per elevation band[3] |

[1] Digitization from historical topographic maps ("Siegfriedkarte") provided by Swisstopo.

[2] Computed ice thickness based on approach by Huss and Farinotti (2012) using glacier outline inventories from Maisch et al. (2000), originally Müller et al. (1976), and from Fischer et al. (2014).

[3] All GIS analyses based on the same digital elevation model (25mx25m) for recent conditions.

Figure A1

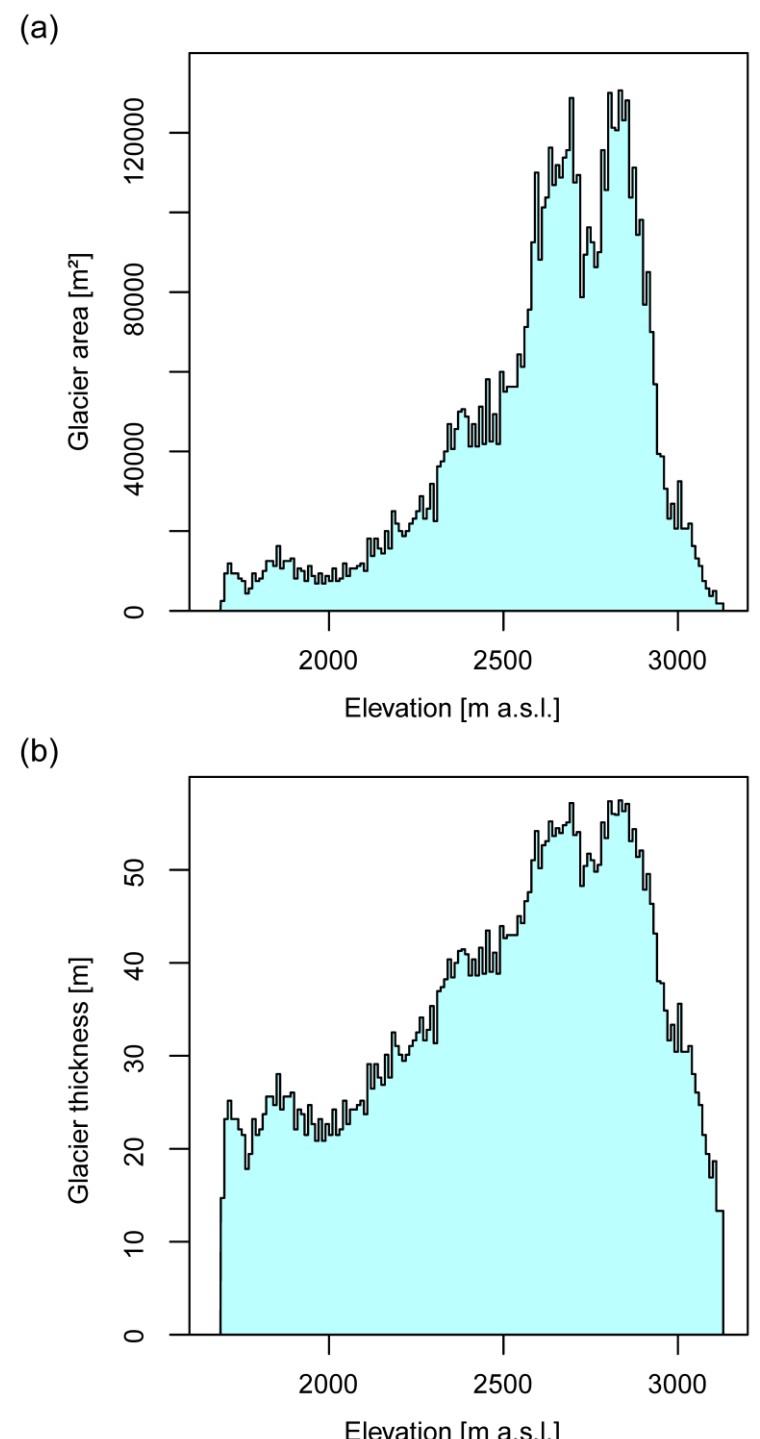

5  Figure A1. Estimated initial glacier geometry as a function of elevation: a) Areal extent. b) Glacier thickness.