# Peer review of "Technical Note: Representing glacier geometry changes in a semidistributed hydrological model"

_Hydrology and Earth System Sciences, 2017_

## Referee Comment (RC1) · Anonymous Referee #1 · 29 Apr 2017

Comments on "Technical Note: Representing glacier dynamics in a semi-distributed hydrological model"

This paper presented a medium-complex glacier melt module and its combination with the HBV model. The idea is good and practically possible. This manuscript will be helpful for beginners of hydrological modelling in glacierized catchments. However, when I try to follow its instructions, I get lost because some critical problems are not clear. And the comparison with previous studies is not well addressed. First of all, the required data should be indicated before the implementation of the model.

Compared to study of Luo et al. (2012), what is the superiority of the $\Delta h$ parameterization method? Why do you choose the $\Delta h$ parameterization method compared to

volume-area scaling?

The statement "The $\Delta$h parameterization method is used to generate the volume-area relationship (Page 4 line 28)" is biased, as the $\Delta$h parameterization is used to generate the spatial distribution of $\Delta$h based on relative elevation.

Page 5 Lines 23 - 31 The relationship of $\Delta$h and mass balance change should be indicated more clearly.

How to account for the snow redistribution effects? Could you provide the detailed information, for example, with equations?

Will the glacier dynamic module take the "equilibrium line altitude" into consideration?

Provide a map to illustrate the study area in the case study. There is only one glacier located within the Alpbach catchment, which is not convincing though $\Delta$h has been successfully incorporated into WASA in Duethmann et al (2015).

Line 6 on Page 4: The sentence "which represents the different albedo of ice compared to snow and typically takes values of about 1 to 2" indicated that glacier melt factor is 1-2 times of snow melt factor. Is it universally true? This need to be verified. How about the debris cover area?

Page 5 line 16: "the conversion of snow to ice takes about 1-3 years" needs references.

Page 9 line 27: I am not clear about how will the glacier retreat. When $\Delta$h is larger than or equal to the thickness of the glacier borders, the glaciers for the corresponding area will disappear?

Page 9 Line 3: this sentence "This is due to the $\Delta$h-approach, which distributes the change in glacier mass balance over the different elevation zones, in combination with the implemented width scaling, which relates a decrease in glacier thickness to a reduction of the glacier area" is confusing. The $\Delta$h-approach is used in combination with the width scaling in this case study? The details should be illustrated in model setup

section.

The idea of classification of "aspect classes" seems to be a very useful conduct? What are the glacio-hydrological effects?

References: Luo, Y., Arnold J., Liu S., Wang X. & Chen X. 2013 Inclusion of glacier processes for distributed hydrological modeling at basin scale with application to a watershed in Tianshan Mountains, northwest China. Journal Of Hydrology 477,72-85.

---

## Author Comment (AC1) · 10 May 2017

We thank the reviewer for her/his valuable comments, which will help us to clarify our technical note. Below we respond (in blue) to the reviewer comments (*in black*)

*This paper presented a medium-complex glacier melt module and its combination with the HBV model. The idea is good and practically possible. This manuscript will be helpful for beginners of hydrological modelling in glacierized catchments.*

We appreciate this positive assessment and actually assume that the paper will also be useful for more advanced modelers, who are faced with the challenge of simulating glacier dynamics in bucket-type hydrological models.

*However, when I try to follow its instructions, I get lost because some critical problems are not clear. And the comparison with previous studies is not well addressed. First of all, the required data should be indicated before the implementation of the model.*

We will clarify the data requirements. Beyond the data needed for the hydrological modeling, i.e., precipitation and temperature and potential evaporation, one needs topographic information and the initial areal extent of the glacier(s) as well as an initial thickness profile.

*Compared to study of Luo et al. (2012), what is the superiority of the Δh parameterization method? Why do you choose the Δh parameterization method compared to volume-area scaling?*

With the volume-area scaling no catchment-specific information is considered, whereas the Δh parameterization allows considering elevation distributions and the thickness profile. In volume-area scaling any volume change directly translates to area changes, although this has not always to be the case (think of two glaciers with the same volume but different thicknesses in their lower part). The Δh parameterization also allows attributing the glacier area changes to the different elevation zones, which would not be directly possible with a simple volume-area scaling, which does not allow to assign the region of glacier shrinkage (see also the discussion in Stahl et al., 2008). As discussed by Huss et al. (2010) the Δh approach is a simple but still physically based approach to consider changing glaciers.

*The statement "The Δh parameterization method is used to generate the volume-area relationship (Page 4 line 28)" is biased, as the Δh parameterization is used to generate the spatial distribution of Δh based on relative elevation.*

We will clarify that we here mean that the Δh approach is used to generate the volume-area relationship for the specific catchment (i.e. the relative elevations from the Huss-equations are transferred to real elevations for this particular setting)

*Page 5 Lines 23 - 31 The relationship of Δh and mass balance change should be indicated more clearly.*

A certain mass balance change over the entire glacier area is distributed to the different elevation zones using the empirical Δh parametrizations from Huss et al. (2010) – see also the detailed description in Huss et al (2010). However, we can include more information if required.

*How to account for the snow redistribution effects?   Could you provide the detailed information, for example, with equations?*

Snow redistribution is an important issue, although not the focus of this technical note. Basically, for higher elevations a threshold was defined  in HBV and all accumulated snow exceeding this threshold is redistributed to lower elevations. This is already described on page 4, lines 17-22. We aware that our approach is a very simple representation, on the other hand,  snow-redistribution also is not the focus of this technical note. Basically we are assuming that the high-altitude non-glacier areas correspond to preferred snow-erosion sites and glaciers and lower-altitude zones correspond to preferred snow transport accumulation sites.

*Will the glacier dynamic module take the "equilibrium line altitude" into consideration?*

This is done implicitly by the Δh approach, but the ELA is not used explicitly

*Provide a map to illustrate the study area in the case study.*

While we agree that maps are in generally helpful, we hesitate to add a map to this technical note. After all the Alpbach catchment is only used as example. However, if required, we could add map like the one below.

[Figure]

*There is only one glacier located within the Alpbach catchment, which is not convincing though Δh has been successfully incorporated into WASA in Duethmann et al (2015).*

The glacier routine presented here was successfully applied to 49 diverse catchments in the Swiss Alps with many of them also having mor than one glacier (see Stahl et al. 2017). We intentionally selected only one catchment with one main glacier for demonstration purposes for this technical note.

Stahl, K. et al. 2017: The snow and glacier melt components of streamflow of the river Rhine and its tributaries considering the influence of climate 30 change, Final report to the International Commission for the Hydrology of the Rhine Basin (CHR). [online] Available from: www.chr-khr.org/en/publications

*Line 6 on Page 4: The sentence "which represents the different albedo of ice compared to snow and typically takes values of about 1 to 2" indicated that glacier melt factor is 1-2 times of snow melt factor. Is it universally true? This need to be verified. How about the debris cover area?*

Thanks for pointing out that we missed to acknowledge the source of this information. Hock (2003) reviewed numerous studies and our factor is based on her Table 1.

Hock, R. (2003). Temperature index melt modelling in mountain areas. Journal of Hydrology, 282, 104–115. http://doi.org/10.1016/S0022-1694(03)00257-9

Debris cover is not an issue in this catchment, but modelers can consider the effect if necessary by adjusting the degree-day factor for ice-melt

*Page 5 line 16: "the conversion of snow to ice takes about 1-3 years" needs references.*

We thank for this questions which made us realize than we were not very exact in our description. What we meant was that it takes 1-3 years until snow is transformed to firn (https://nsidc.org/cryosphere/glaciers/questions/formed.html). The further transformation to ice varies with climatic conditions and can take 10 to more than 100 years. However, when simulating melt rates, we here treated the firn as ice when simulating melt rates. One can also note that this approach and even the used parameter value agrees with the approach used fir the snow-firn(ice) conversion by Luo et al. (2013).

*Page 9 line 27: I am not clear about how will the glacier retreat. When Δh is larger than or equal to the thickness of the glacier borders, the glaciers for the corresponding area will disappear?*

Yes exactly, as described in the last paragraph on page 5 (following Eq. 6): if the computed change in an elevation interval is larger than the remaining ice, the glacier will be reduced by the corresponding area. In addition the glacier area will also be (slightly) reduced for higher elevations when the thickness is decreasing (Eq. 7).

*Page 9 Line 3: this sentence "This is due to the Δh-approach, which distributes the change in glacier mass balance over the different elevation zones, in combination with the implemented width scaling, which relates a decrease in glacier thickness to a reduction of the glacier area" is confusing. The Δh-approach is used in combination with the width scaling in this case study? The details should be illustrated in model setup section.*

We agree that this is a bit confusing and will try to clarify this in the revision. While in the original Δh-approach the glacier area is reduced only from the lowest elevations, we added an area reduction at higher elevations when the thickness in these zones decreased to improve the realism of the glacier area changes.

*The idea of classification of "aspect classes" seems to be a very useful conduct? What are the glacio-hydrological effects?*

The use of aspect classes will cause the ice and snow melt to be faster or slower and, taken together, to spread over a longer time period. This approach has been used previously, for instance, in the ETH version of the HBV model (e.g., Hottelet et al., 1993; Hagg et al., 2007).

Hottelet, C., Braun, L. N., Leibundgut, C. and Rieg, A.: Simulation of Snowpack and Discharge in an Alpine Karst Basin, in Snow and Glacier Hydrology (Proceedings of the Kathmandu Symposium, November 1992). IAHSPubl.no. 218, pp. 249–260., 1993.

Hagg, W., Braun, L. N., Kuhn, M. and Nesgaard, T. I.: Modelling of hydrological response to climate change in glacierized Central Asian catchments, J. Hydrol., 332(1–2), 40–53, doi:10.1016/j.jhydrol.2006.06.021, 2007.

*References: Luo, Y., Arnold J., Liu S., Wang X. & Chen X. 2013 Inclusion of glacier processes for distributed hydrological modeling at basin scale with application to a watershed in Tianshan Mountains, northwest China. Journal Of Hydrology 477,72-85.*

We thank the reviewer for making us aware of this interesting paper.

---

## Referee Comment (RC2) · Anonymous Referee #2 · 19 May 2017

This paper describes the implementation of a simple approach to calculate glacier geometry change and retreat designed for hydrological models. This approach, the dh-parameterization, has previously been published in HESS by Huss et al., 2010 ("Future high-mountain hydrology: a new parameterization of glacier retreat"). This parameterization has already been included in various glacio-hydrological models (e.g. Duethmann et al., 2015; Li et al., 2015), its performance has been evaluated in several studies (e.g. Vincent et al. 2014, Huss et al., 2014), and has been used to calculate the evolution of all glaciers globally related to sea-level change assessments (Huss and Hock, 2015). The present paper describes an implementation of the dh-parameterization into the framework of the HBV-light model, including an example application in the Swiss

[Figure]

Alps.

The paper is well written and clear in most places. However, some partly important issues need to be resolved before it can be recommended for publication:

Novelty: I am a little bit concerned about the novelty of the study. The paper describes the implementation of a published approach developed for hydrological modelling into another model. Differences to the original implementation are small. The authors clearly describe the origin of the approach and make complete reference to it. For increasing the justification of publishing this article, however, the authors might try to better point out where their paper goes beyond the original study of the dh-parameterization and where the present description facilitates the application by hydrological modellers. The performance of the approach is not extensively tested so far and also the implementation of a glacier advance scheme has been implemented for the dh-parameterization by a different study (referenced in the manuscript). Nevertheless, I think there are some drawbacks to previous implementations / descriptions of the parameterization that could be more strongly highlighted in this paper: (1) How well does the glacier advance module perform? (2) How to implement the glacier retreat model if no ice thickness data are readily available? Some strategies must be provided to make the approach useful to the hydrological community (see also next comment). (3) How do the different implementations of the parameterization affect runoff (i.e. what error in runoff is committed when glacier retreat is not or insufficiently taken into account? Although (1) and (3) are somehow covered in Figure 3 the discussion is completely qualitative. The errors and their significance in comparison to the measurement uncertainties should be stated.

Ice thickness: One of the most important drawbacks of a straight-forward implementation of the dh-parameterization is the need for data on glacier ice thickness distribution. Whereas several approaches to estimate ice thickness with glaciological models have been developed in the last years (see Farinotti et al., 2017, The Cryosphere, for an overview) many hydrological modellers will not have direct access to ice thick-

ness data for their study site in the desired spatial resolution etc. The present study benefits from a data set directly provided externally by the developer of the original dh-parameterization. The present study aims at describing the implementation of the dh-approach into simple hydrological models: Without the availability of ice thickness data this is however not possible – this data is the bottleneck for the dh-parameterization! In my opinion, more effort should be invested in this paper to also describe simple strategies to overcome this restriction. Furthermore, this issue also needs to be much more prominently mentioned in the introduction and the method description. For most of the time the reader is left with no clear idea where the ice thickness is information is taken from – it just seems to be available.

Mass conservation: The dh-parameterization aims at being mass conserving which is crucial for hydrological modelling. In many implementations of the dh-parameterization, mass conservation is a critical issue and can be violated if it is not explicitly ensured. The authors should check if mass is conserved in their implementation and describe their strategy to ensure mass conservation.

Different glaciers: It is unclear what happens if different (separated) glaciers are present in the catchment. Can the authors' implementation of the parameterization only be applied to catchments that contain one glacier? What are the limitations when several glaciers are present in the catchment?

Model calibration and validation: HBV-light is applied for an Alpine catchment for a period of >100 years. It remains unclear in the present paper how the model was calibrated and validated for this application. Some more details are necessary.

Impact on runoff: see also comment above. Here, the present study using a simple and operational hydrological model could go one step further than previous studies: What is the effect of using the glacier retreat parameterization on calculated runoff? Is it possible to quantify the benefit?

Detailed comments:

Page 1, line 34: Some references should be provided here

Page 2, line 2: Actually, full hydrological models, incorporating glacier dynamics explicitly, have been published in the last years (e.g. Naz et al., 2014, HESS; Frans et al., 2016, HP). Reference to these approaches should be made, also to justify the use of strongly simplified glacier models.

Page 3, line 21: ice accumulation => snow accumulation

page 4, line 16: A transformation time of 1-3 years is too fast. Please provide a reference and choose more realistic numbers

page 4, line 17-22: The description of snow redistribution is unclear and needs revision. There seems to be quite arbitrary choices in this approach and justification is required.

Page 4, line 26: "single-valued relation between glacier mass balance and glacier area". Is this really the case? This does not make sense in my opinion and also seems to be inconsistent with the argumentation in the paper. Has the word "area CHANGE" been lost? But even then, the dh-parameeerization should be prescribe such a single-valued relation.

Page 4, line 31: Here, and elsewhere. I do not like the partly very method-specific descriptions. Of course the implementation in the HBV-light model relies on a so-called "glacier profile" file. But the paper aims at providing a methodological description for implementing a glacier retreat model. So, I would avoid notions that are too specific to the authors' own model.

Page 5, line 29: Where is h_i,old taken from? (see also general comment above)

page 6, line 35: Please provide a reference for glacier area in 2010 and a more accurate number (i.e. 1-2 digits).

Page 7, line 26: Where is glacier surface geometry for the year 1900 taken from?

Page 8, top: I suggest having a kind of data section here to better organize the input

data for the example catchment

page 8, line 22: It is not clear where the initial distribution of ice thickness around 1900 is coming from.

Page 8, line 29: What is done here exactly? It seems that in addition to the dh-parameterization also volume-area scaling has been used. Please describe how and why. I strongly suggest to not combine volume-area scaling and the dh-parameterization. These are separate approaches that conceptually do not go together well

Page 8, line 31: better 900 kg m-3

page 9, line 20: Instead of using only glacier areas for model validation, the change in glacier volume would be a much better measure to assess model performance in terms of discharge. Such data would be available for the investigated catchment based on Fischer et al. (2015, The Cryosphere).

Page 11, line 16: Well, as the authors describe in the introduction, this approach already has been implemented in other hydrological models. These sentences should be reformulated to better reflect this.

---

## Author Comment (AC2) · 8 Jun 2017

We thank the reviewer for her/his in general positive assessment of our manuscript and the helpful comments. Below we respond (in blue) to the reviewer comments (*in black*)

*The paper is well written and clear in most places.*
Thanks!

*Novelty: I am a little bit concerned about the novelty of the study. The paper describes the implementation of a published approach developed for hydrological modelling into another model. Differences to the original implementation are small. The authors clearly describe the origin of the approach and make complete reference to it. For increasing the justification of publishing this article, however, the authors might try to better point out where their paper goes beyond the original study of the dh-parameterization and where the present description facilitates the application by hydrological modellers.*
We find submission of the manuscript as a Technical Note to be justified as, when we implemented the method, we indeed found most previous descriptions of other implementations to be not very detailed (= reproducible). While the method itself is not new, the comparison of the different implementations of the ∆h-parameterization into a widely used semi-distributed hydrological catchment model and the test of their effect over a >100 year simulation period is novel, in particular with the use of a look-up table that is generated by a pre-simulation application of the ∆h-parameterization to allow the advancement of the glacier.

*The performance of the approach is not extensively tested so far and also the implementation of a glacier advance scheme has been implemented for the dh-parameterization by a different study (referenced in the manuscript). Nevertheless, I think there are some drawbacks to previous implementations / descriptions of the parameterization that could be more strongly highlighted in this paper: (1) How well does the glacier advance module perform?*
We agree that a detailed evaluation of model performance would be beneficial, even if the ∆h-parameterization itself is well-established (not novel as correctly stated above). However, this test would require data from several glaciers and the analyses would go beyond the scope of the technical note. We want to emphasize that we on purpose decided to present the new implementation as technical note and not as full paper. We have made this new model version freely available to other researchers and first studies using this new routine are appearing in literature (e.g., Van Tiel, M., Teuling, A. J., Wanders, N., Vis, M. J. P., Stahl, K., and Van Loon, A. F.: The role of glacier dynamics and threshold definition in the characterisation of future streamflow droughts in glacierised catchments, Hydrol. Earth Syst. Sci. Discuss., doi:10.5194/hess-2017-119, in review, 2017). For these studies it is important, that the model implementation is clearly described in literature and these studies will then ultimately contribute to assessing the model performances in various environments (beyond what would be able for us alone to achieve).

*(2) How to implement the glacier retreat model if no ice thickness data are readily available? Some strategies must be provided to make the approach useful to the hydrological community (see also next comment).*
*Ice thickness: One of the most important drawbacks of a straight-forward implementation of the dh-parameterization is the need for data on glacier ice thickness distribution. Whereas several approaches to estimate ice thickness with glaciological models have been developed in the last years (see Farinotti et al., 2017, The Cryosphere, for an overview) many hydrological modellers will not have direct access to ice thick-*

*ness data for their study site in the desired spatial resolution etc. The present study benefits from a data set directly provided externally by the developer of the original dh-parameterization. The present study aims at describing the implementation of the dh-approach into simple hydrological models: Without the availability of ice thickness data this is however not possible – this data is the bottleneck for the dh-parameterization! In my opinion, more effort should be invested in this paper to also describe simple strategies to overcome this restriction. Furthermore, this issue also needs to be much more prominently mentioned in the introduction and the method description. For most of the time the reader is left with no clear idea where the ice thickness is information is taken from – it just seems to be available.*

We fully agree that obtaining data on initial glacier ice thickness distribution is a challenge. However, glaciologists have developed various methods for this as nicely reviewed by Farinotti et al, 2017, and, more important, this challenge applies to any approach where glacier area is considered to change. So, unless we want to consider glaciers as static in area (i.e., infinite ice thickness) there is actually no way to get around the initial ice thickness estimation. We will clarify better where our initial ice thickness came from, how a glacier profile is defined in the specific implementation and the assumptions made, but for alternatives we think adding a reference to Farinotti et al. 2017 will be more beneficial than some (arbitrary) testing of other options.

*(3) How do the different implementations of the parameterization affect runoff (i.e. what error in runoff is committed when glacier retreat is not or insufficiently taken into account? Although (1) and (3) are somehow covered in Figure 3 the discussion is completely qualitative. The errors and their significance in comparison to the measurement uncertainties should be stated.*

We do show parameterization uncertainties in the figures. The ranges can be complemented with some numbers in the text if it's generally felt that this is needed. Annual rather than cumulative flow in Figure 3 may also provide an easier to interpret values. We suggest to discuss this more clearly in the revised version. Of course the significance for the simulation of total runoff depends strongly on the catchment (i.e., its glacier coverage). However, even if in some catchments the quantitative effect on simulated runoff would be relatively small, an adequate representation of changing glacier area is definitely desirable because it enables additional model validation (in terms of glacier simulation) and will help in the identification of the most appropriate snow/ice related model parameters being crucial for modeling alpine catchments.

*Mass conservation: The dh-parameterization aims at being mass conserving which is crucial for hydrological modelling. In many implementations of the dh-parameterization, mass conservation is a critical issue and can be violated if it is not explicitly ensured. The authors should check if mass is conserved in their implementation and describe their strategy to ensure mass conservation.*

We did ensure that volume, area and thickness are related at all times in our application and match the modelled the mass balance. This is an important comment and indeed this was one of the reasons that prompted us to introduce the width scaling and to ensure mass conservation including a redistribution of water in model stores when some glacier elevation zones melt out and change landcover type (actually, we now realized that this technical detail needs to be mentioned in revised manuscript). We will explain this more explicitly in the revised version.

*Different glaciers: It is unclear what happens if different (separated) glaciers are present in the catchment. Can the authors' implementation of the parameterization only be applied to catchments that contain one glacier? What are the limitations when*

*several glaciers are present in the catchment?*

If there are several glaciers within one (sub)catchment these are represented in a summarized way, i.e. as one glacier. But the model could, in principle, be extended to include several glaciers or the model could be applied to individual glaciers separately.

*Model calibration and validation: HBV-light is applied for an Alpine catchment for a period of >100 years. It remains unclear in the present paper how the model was calibrated and validated for this application. Some more details are necessary.*

Calibration and validation are described in detail by Stahl et al. (2017) and we will add some more information on this in the revision.

Impact on runoff: see also comment above. Here, the present study using a simple and operational hydrological model could go one step further than previous studies: What is the effect of using the glacier retreat parameterization on calculated runoff? Is it possible to quantify the benefit?

This is an important aspect – also see our responses above: we can add some summary numbers to complement the figures in the revised version. The effect of considering changing glacier areas on runoff differs of course depending on catchment and glacier area change. We demonstrate the effect with an example here, but more examples are provided by Stahl et al., 2017. The new routine will potentially be even more important for the simulation of future scenarios beyond 'peak water'.

*Detailed comments:*
*Page 1, line 34: Some references should be provided here*
We will add these
*Page 2, line 2: Actually, full hydrological models, incorporating glacier dynamics explic-itly, have been published in the last years (e.g. Naz et al., 2014, HESS; Frans et al., 2016, HP). Reference to these approaches should be made, also to justify the use of strongly simplified glacier models.*
Thanks for making us aware of these studies, which we will include in the revised version.

*Page 3, line 21: ice accumulation => snow accumulation*
We will change this to Snow and ice accumulation

*page 4, line 16: A transformation time of 1-3 years is too fast. Please provide a refer-ence and choose more realistic numbers*
We agree. What we meant was that it takes 1-3 years until snow is transformed to firn (https://nsidc.org/cryosphere/glaciers/questions/formed.html). The further transformation to ice varies with climatic conditions and can take 10 to more than 100 years. However, when simulating melt rates, we here treated the firn as ice when simulating melt rates. One can also note that this approach and even the used parameter value agree with the approach used for the snow-firn(ice) conversion by Luo et al. (2013 in Journal of Hydrology).

*page 4, line 17-22: The description of snow redistribution is unclear and needs revision. There seems to be quite arbitrary choices in this approach and justification is required.*
We will expand the description, however, this aspect is not the focus of this manuscript. Snow redistribution is a challenge on its own, which is further discussed in Freudiger et al. (WIRES Water, in press – reference will be added).

*Page 4, line 26: "single-valued relation between glacier mass balance and glacier area". Is this really the case? This does not make sense in my opinion and also seems to be inconsistent with the argumentation in the paper. Has the word "area CHANGE" been lost? But even then, the dh-parameerization should be prescribe such a single-valued relation.*

Thanks. In the revision we will delete the word balance, i.e. relation between glacier mass and area. Yes, the application of the Δh-parameterization leads to such a single-valued relation between glacier mass and glacier area in each of the different elevation zones (as stated in the same paragraph lines 28/29). We will clarify this part.

*Page 4, line 31: Here, and elsewhere. I do not like the partly very method-specific descriptions. Of course the implementation in the HBV-light model relies on a so-called "glacier profile" file. But the paper aims at providing a methodological description for implementing a glacier retreat model. So, I would avoid notions that are too specific to the authors' own model.*

We both agree and disagree. The issue is that we need to provide enough details. Please note that we on purpose did not mention things like file names etc.

*Page 5, line 29: Where is h_i,old taken from? (see also general comment above)*

h_i,old is computed iteratively, for initial value see above

*page 6, line 35: Please provide a reference for glacier area in 2010 and a more accurate number (i.e. 1-2 digits).*

We will provide another digit and reference. The estimates are 4.03 km$^2$ for 2010 (Huss pers. comm., based on Fischer et al. 2014) and 4.57 km² based on Paul et al. 2011

*Page 7, line 26: Where is glacier surface geometry for the year 1900 taken from?*

This is described further down (from line 26) and will be extended in the revised version.

*Page 8, top: I suggest having a kind of data section here to better organize the input data for the example catchment*

We will extend the description of the input data.

*page 8, line 22: It is not clear where the initial distribution of ice thickness around 1900 is coming from.*

See above, will be clarified

*Page 8, line 29: What is done here exactly? It seems that in addition to the dh-parameterization also volume-area scaling has been used. Please describe how and why.*
*I strongly suggest to not combine volume-area scaling and the dh-parameterization. These are separate approaches that conceptually do not go together well*

This text does not refer to the volume-area scaling implemented in the glacier routine but to the initial conditions. However, it seems that the reviewer interpreted it that former way. We, thus, realize that there are two issues:
1)      We will clarify that page 8 line 29 is about the reconstruction of the initial ice thickness distribution. This text refers to the reconstruction of the initial ice thickness which is described in Stahl et al. (2017) and for which we will also provide a more detailed explanation in the revised version (see

comment above). Basically our reconstruction is mainly based on two physically-based relationships taken from Bahr et al. (1997): i) the general volume-area scaling relation ($V= c \cdot A^{\gamma}$ with $\gamma = 1.375$) and ii) a proportionality of glacier width and the square root of glacier thickness. For clarification: While the latter relation is indeed also used in our model approach to represent the change of glacier width within a certain elevation band (as explained on page 6 line 5 ff. and also applied by Huss & Hock 2015), the "classical" glacier volume-area scaling is not directly used in the modelling itself. Hence, it is not used in combination with the $\Delta$h-parameterization in the implementation of our glacier dynamic representation approach as perhaps suspected. It was solely used to derive estimates on total glacier volume for the state in the years ~1900, ~1940, and 2003. These obtained glacier volume estimates were used as data for model initialization (initial ice thickness distribution) and calibration (see observation based data in Figure 3 b).

2)      Regarding combining volume-area scaling with the $\Delta$h-parameterization, we argue that the original Huss approach is somewhat unrealistic as the glacier area only changes at the glacier terminus. The scaling approach allows considering the fact that a thinning glacier in some elevation zone also causes (small) decreases in area. We do not see a fundamental problem of the use of a scaling approach to allow for these (small) areal changes.

*Page 8, line 31: better 900 kg m-3*
Will be changed

*page 9, line 20: Instead of using only glacier areas for model validation, the change in glacier volume would be a much better measure to assess model performance in terms of discharge. Such data would be available for the investigated catchment based on Fischer et al. (2015, The Cryosphere).*
This is an interesting suggestion. However, please note that the data from Fischer et al. (2015) covers only the period 1980-2010, which makes such a validation difficult as the initial conditions might not match. Therefore this option might be less suitable after all, but we will consider it as an additional model validation. However, please note that Figure 3 b  (simulated change in glacier water equivalent) already visualizes the performance in terms of change in total glacier volume. The shown observation based change in glacier ice water equivalent values for the years 1940, 1973, and 2003, which were also used for model calibration, actually are directly converted from our estimates on glacier volume based on glacier area data for those years. We agree that for hydrological modeling an accurate simulation of glacier volume matters more than that of glacier area. That is why we used glacier volume changes (as glacier water equivalent changes) for model calibration. We think an additional validation using the data by Fischer et al. (2015) might be mainly informative regarding the uncertainty of the glacier volume estimates used by us (differences compared to reference data by Fischer et al.) rather than regarding the capability of the model to represent the change in glacier volume adequately.

*Page 11, line 16: Well, as the authors describe in the introduction, this approach already has been implemented in other hydrological models. These sentences should be reformulated to better reflect this.*
We were apparently not clear what 'this approach' is referring to. Here we meant our approach (specific implementation to represent glacier dynamics in a semi-distributed hydrological model), whereas the reviewers' comment refers to the dh-parameterization. We will clarify this.

---

## Author Response (AR1)

We thank the editor and both reviewers for their valuable comments, which helped us to substantially improve our technical note. Below we respond (in red) to the editor/reviewer comments (*in black*)

***Editor***

*I'm sorry for the delays in this processing, a number of personal issues have come up. I find this very difficult to make an opinion on this technical note for 2 core reasons. Firstly a technical note for HESS is detailed as thus:*

*Technical notes report new developments, significant advances, and novel aspects of experimental and theoretical methods and techniques which are relevant for scientific investigations within the journal scope. Manuscripts of this type should be short (a few pages only). Highly detailed and specific technical information such as computer programme code or user manuals can be included as electronic supplements. The manuscript title must start with "Technical note:". For manuscripts focused on the development and description of numerical models and model components, we recommend submission to the EGU interactive open-access journal Geoscientific Model Development (GMD).*

*Whilst to some extent the authors acknowledge a lack of 'novelty' due to previous use of these formulations. I'm going to have to see a better justification in the final response to say why this is a standalone technical note that fits the comments about HESS technical notes detailed above. There are comments made to the extent that a paper was justified to enable the model to be 'reproducible'. I'm sorry but I then do not see enough technical depth (many which the reviewers raise) to make this so. So again more work is needed here to justify publication.*

We agree that the novelty aspects are limited, but want to stress the point that our approach allows to use the $\Delta h$ parameterization by Huss et al. (2010) approach for glaciers which advance during some time. This has not been feasible before. The main contribution of our TN, however, is to show how the method can be implemented into a hydrological model. In order to make hydrological modeling reproducible, the clear documentation of model routines is important. We realized that our descriptions needed to be clarified to reach this goal. For this we have significantly rewritten the manuscript and added a flowchart as new figure.

*Secondly the authors responses to some very good critical analyses are very general. The author response is meant to clarify what changes will be made to answer the points raised by reviewers. That is often very limited and I have no clear idea as to what the new paper would look like at this moment in time.*

We are sorry for the partly too general responses in the discussion phase. We hope that our responses below and the concrete changes make things clearer.

***So recuasse of these two points then i will need to confirm publication once a clearer development of the paper and revisions is realised and more of a clear justification as to why a technical note is justified. Either due to novelty or to explain to the audience how this helps researchers directly***

***implement these methods and resolves some of the practical aspects in doing so…. Also the reviewers did request seeing the revised manuscript again, Jim***

We are looking forward to the feedbacks of the editor and the reviewers on our revised version. We feel that this review-revision round substantially improved the clarity of our manuscript.

*Reviewer #1*

*This paper presented a medium-complex glacier melt module and its combination with the HBV model. The idea is good and practically possible.  This manuscript will be helpful for beginners of hydrological modelling in glacierized catchments.*

We appreciate this positive assessment and actually assume that the paper will also be useful for more advanced modelers, who are faced with the challenge of simulating glacier dynamics in bucket-type hydrological models.

*However, when I try to follow its instructions, I get lost because some critical problems are not clear. And the comparison with previous studies is not well addressed. First of all, the required data should be indicated before the implementation of the model.*

We clarified the data requirements. Beyond the data needed for the hydrological modeling, i.e., precipitation and temperature and potential evaporation, one needs the initial areal extent of the glacier(s) as well as an initial thickness profile, i.e. a table with glacier area and ice thickness values for each elevation band.

*Compared to study of Luo et al.  (2012), what is the superiority of the Δh parameterization method? Why do you choose the Δh parameterization method compared to volume-area scaling?*

With the volume-area scaling no catchment-specific information is considered, whereas the Δh parameterization allows considering elevation distributions and the thickness profile. In volume-area scaling any volume change directly translates to area changes, although this has not always to be the case. The Δh parameterization allows attributing the glacier area changes to the different elevation zones, which would not be directly possible with a simple volume-area scaling, which does not allow assigning the region of glacier shrinkage to any particular elevation zone (see also the discussion in Stahl et al., 2008). As discussed by Huss et al. (2010) the Δh approach is a simple but still physically based approach to consider changing glaciers. We added this motivation to the introduction text. We added a discussion of these advantages of the *Δh*-parameterization compared to volume-area scaling in the revised manuscript.

*The statement "The Δh parameterization method is used to generate the volume-area relationship (Page 4 line 28)" is biased, as the Δh parameterization is used to generate the spatial distribution of Δh based on relative elevation.*

We clarified that we here mean that the Δh approach is used to generate the volume-area relationship for the specific catchment (i.e. the relative elevations from the Huss-equations are transferred to real elevations for this particular setting)

*Page 5 Lines 23 - 31 The relationship of Δh and mass balance change should be indicated more clearly.*

Each annual net mass change over the entire glacier area in a catchment calculated from the semi-distributed mass balance routine is distributed to the different elevation zones using the empirical Δh parametrizations from Huss et al. (2010) – see also the detailed description in Huss et al (2010). We clarified this in the text by changing the notation ('k+1' and 'k' instead of 'old' and 'new')

*How to account for the snow redistribution effects?   Could you provide the detailed information, for example, with equations?*

Snow redistribution is an important issue, although not the focus of this technical note. Basically, for higher elevations a threshold was defined in HBV and all accumulated snow exceeding this threshold is redistributed to lower elevations. This was already described on page 4, lines 17–22  (now on page 5 lines 12ff slightly revised). We are aware that our approach is a very simplified representation, on the other hand, snow-redistribution also is not the focus of this technical note. Basically we are assuming that the high-altitude non-glacier areas correspond to preferred snow-erosion sites and glaciers and lower-altitude zones correspond to preferred snow deposition sites. We also added a reference to a new review paper focusing explicitly on snow redistribution (Freudiger et al., 2017).

*Will the glacier dynamic module take the "equilibrium line altitude" into consideration?*

This is done implicitly by the Δh approach, but the ELA is not used explicitly

*Provide a map to illustrate the study area in the case study.*

While we agree that maps are in generally helpful, we hesitate to add a map to this technical note. After all the Alpbach catchment is only used as example. However, we added the map below in our HESS-D response. For the paper we would prefer not to add a map, but if the editor/reviewers wish, we could add such a map in the final manuscript.

[Figure]

Legend:
- ▼ Streamflow gauge
- Glacier cover 2003
- Glacier cover 1973
- Glacier cover ~1940
- Glacier cover ~1900

*There is only one glacier located within the Alpbach catchment, which is not convincing though Δh has been successfully incorporated into WASA in Duethmann et al (2015).*

The glacier routine presented here was successfully applied to 49 diverse catchments in the Swiss Alps with many of them also having more than one glacier (see Stahl et al. 2017). We intentionally selected only one catchment with one main glacier for demonstration purposes for this technical note.

Stahl, K. et al. 2017: The snow and glacier melt components of streamflow of the river Rhine and its tributaries considering the influence of climate , Final report to the International Commission for the Hydrology of the Rhine Basin (CHR). [online] Available from: www.chr-khr.org/en/publications

We added this to the text.

*Line 6 on Page 4: The sentence "which represents the different albedo of ice compared to snow and typically takes values of about 1 to 2" indicated that glacier melt factor is 1-2 times of snow melt factor. Is it universally true? This need to be verified. How about the debris cover area?*

Thanks for pointing out that we missed to acknowledge the source of this information. Hock (2003) reviewed numerous studies and our factor is based on her Table 1. We added this reference to the text.

Hock, R. (2003). Temperature index melt modelling in mountain areas. Journal of Hydrology, 282, 104–115. http://doi.org/10.1016/S0022-1694(03)00257-9

Debris cover is not an issue in this catchment, but modelers can consider the effect if necessary by adjusting the degree-day factor for ice melt

*Page 5 line 16: "the conversion of snow to ice takes about 1-3 years" needs references.*

We thank for this comment which made us realize that we were not very exact in our description. What we meant was that it takes 1-3 years until snow is transformed to firn (https://nsidc.org/cryosphere/glaciers/questions/formed.html). The further transformation to ice varies with climatic conditions and can take 10 to more than 100 years. However, when simulating melt rates, we here treated the firn as ice when simulating melt rates. One can also note that this approach and even the used parameter value agrees with the approach used for the snow-firn(ice) conversion by Luo et al. (2013).

We clarified this in the text.

*Page 9 line 27: I am not clear about how will the glacier retreat. When Δh is larger than or equal to the thickness of the glacier borders, the glaciers for the corresponding area will disappear?*

Yes exactly: if the computed change in an elevation interval is larger than the remaining ice, the glacier area in this interval is set to zero and the model glacier will be reduced by the corresponding area. In addition the glacier area will also be (slightly) reduced for higher elevation intervals when the thickness is decreasing (Eq 7).

*Page 9 Line 3: this sentence "This is due to the Δh-approach, which distributes the change in glacier mass balance over the different elevation zones, in combination with the implemented width scaling, which relates a decrease in glacier thickness to a reduction of the glacier area" is confusing. The Δh-approach is used in combination with the width scaling in this case study? The details should be illustrated in model setup section.*

While in the original Δh-approach the glacier area is reduced only from the lowest elevations, we added an area reduction at higher elevations when the thickness in these zones decreased to improve the realism of the glacier area changes. We amended this sentence in the revised version to be more specific. The combination of the *Δh-* approach with the width scaling is shown in the new Figure 1 and described in detail in the model section 2.1 (assuming that the reviewer with "model setup section" refers to that section?) in the paragraph before Eq. 7. Figure 2c illustrates the glacier area over elevation bands that results from this combined application of the *Δh*-parametrization and the width scaling according to Eq. 7, whereas the illustration of simulated glacier volume over elevation is not affected by the width scaling, i.e. results from the application of the original *Δh*-parametrization method without

the glacier width scaling. We tried to clarify this by adding some more explanations and additional references to the illustrations in Figure 1 in the revised manuscript.

*The idea of classification of "aspect classes" seems to be a very useful conduct? What are the glacio-hydrological effects?*

The use of aspect classes will cause the ice and snow melt to be faster or slower and, taken together, to spread over a longer time period. This approach has been used previously, for instance, in the ETH version of the HBV model (e.g., Hottelet et al., 1993; Hagg et al., 2007).

We added this to the text.

Hottelet, C., Braun, L. N., Leibundgut, C. and Rieg, A.: Simulation of Snowpack and Discharge in an Alpine Karst Basin, in Snow and Glacier Hydrology (Proceedings of the Kathmandu Symposium, November 1992). IAHS Publ.no. 218, pp. 249–260., 1993.

Hagg, W., Braun, L. N., Kuhn, M. and Nesgaard, T. I.: Modelling of hydrological response to climate change in glacierized Central Asian catchments, J. Hydrol., 332(1–2), 40–53, doi:10.1016/j.jhydrol.2006.06.021, 2007.

*References: Luo, Y., Arnold J., Liu S., Wang X. & Chen X. 2013 Inclusion of glacier processes for distributed hydrological modeling at basin scale with application to a watershed in Tianshan Mountains, northwest China. Journal Of Hydrology 477,72-85.*

We thank the reviewer for making us aware of this interesting paper, to which we refer now in the revised manuscript.

**Reviewer #2**

*The paper is well written and clear in most places.*
Thanks!

*Novelty: I am a little bit concerned about the novelty of the study. The paper describes the implementation of a published approach developed for hydrological modelling into another model. Differences to the original implementation are small. The authors clearly describe the origin of the approach and make complete reference to it. For increasing the justification of publishing this article, however, the authors might try to better point out where their paper goes beyond the original study of the dh-parameterization and where the present description facilitates the application by hydrological modellers.*
We indeed found most previous descriptions of other implementations to be not detailed to be reproducible and hence find a Technical Note on this implementation warranted. While the method itself is not new, two novel aspects are (i) the comparison of the different implementations of the Δh-parameterization into a widely used semi-distributed hydrological catchment model and the test of their effect over a >100 year simulation period, and (ii) the use of a look-up table that is generated by a pre-simulation application of the Δh-parameterization to allow the advancement of the glacier. We further describe in detail amendments such as the width scaling and the re-distribution of the 'model water'

after changing the landcover of a particular model unit from glacier to non-glacier to provide a complete description of the implementation (see also new Figure 1)

*The performance of the approach is not extensively tested so far and also the implementation of a glacier advance scheme has been implemented for the dh-parameterization by a different study (referenced in the manuscript). Nevertheless, I think there are some drawbacks to previous implementations / descriptions of the parameterization that could be more strongly highlighted in this paper: (1) How well does the glacier advance module perform?*

We agree that a detailed evaluation of model performance would be beneficial, even if the Δh-parameterization itself is well-established (not novel as correctly stated above). However, this test would require data from several glaciers and the analyses would go beyond the scope of the technical note. Actually there is a recent paper by Huss and Hock (2015) with a global assessment of the Δh-parameterization which we now refer to in our manuscript. We want to emphasize that we on purpose decided to present the new implementation as technical note and not as full paper. We have made this new model version freely available to other researchers and first studies using this new routine are appearing in literature (e.g., Van Tiel, M., Teuling, A. J., Wanders, N., Vis, M. J. P., Stahl, K., and Van Loon, A. F.: The role of glacier dynamics and threshold definition in the characterisation of future streamflow droughts in glacierised catchments, Hydrol. Earth Syst. Sci. Discuss., doi:10.5194/hess-2017-119, in review, 2017, or Etter, Simon, Nans Addor, Matthias Huss, David Finger, Climate change impacts on future snow, ice and rain runoff in a Swiss mountain catchment using multi-dataset calibration, Journal of Hydrology: Regional Studies, Volume 13, October 2017, Pages 222-239, ISSN 2214-5818, https://doi.org/10.1016/j.ejrh.2017.08.005.). With this technical note we want to provide a  clear description of the model implementation applied in these and future studies which will then ultimately contribute to assessing the model performances in various environments (beyond what would be able for us alone to achieve).

*(2) How to implement the glacier retreat model if no ice thickness data are readily available? Some strategies must be provided to make the approach useful to the hydrological community (see also next comment).*

*Ice thickness: One of the most important drawbacks of a straight-forward implementation of the dh-parameterization is the need for data on glacier ice thickness distribution. Whereas several approaches to estimate ice thickness with glaciological models have been developed in the last years (see Farinotti et al., 2017, The Cryosphere, for an overview) many hydrological modellers will not have direct access to ice thickness data for their study site in the desired spatial resolution etc. The present study benefits from a data set directly provided externally by the developer of the original dh-parameterization. The present study aims at describing the implementation of the dh-approach into simple hydrological models: Without the availability of ice thickness data this is however not possible – this data is the bottleneck for the dh-parameterization! In my opinion, more effort should be invested in this paper to also describe simple strategies to overcome this restriction. Furthermore, this issue also needs to be much more prominently mentioned in the introduction and the method description. For most of the time the reader is left with no clear idea where the ice thickness is information is taken from – it just seems to be available.*

We fully agree that obtaining data on initial glacier ice thickness distribution is a challengeHowever, glaciologists have developed various methods for this as nicely reviewed by Farinotti et al, 2017, and, more important, this challenge applies to any approach where the model representation of glacier area

is considered to change. So, unless we want to consider glaciers as static in area (i.e., infinite ice thickness) there is actually no way to get around the initial ice thickness estimation. We now added an appendix and explain better where our initial ice thickness came from, how a glacier profile is defined in the specific implementation and which assumptions were made.  We benefited from ice thickness data provided by Matthias Huss (generated based on approaches included in Farinotti et al. 2017). However, the special challenge with our example was obtaining ice thickness distributions for the state a century back. For simulations starting more recent, either ice thickness data sources are available or appropriate data to apply methods described in Farinotti et al. 2017 (we added this valuable review as a reference)

*(3) How do the different implementations of the parameterization affect*
*runoff (i.e. what error in runoff is committed when glacier retreat is not or insufficiently*
*taken into account? Although (1) and (3) are somehow covered in Figure 3 the discus-*
*sion is completely qualitative. The errors and their significance in comparison to the*
*measurement uncertainties should be stated.*
We actually do show parameterization uncertainties in the figures (ranges in Figure 4 (3 in previous version)). Of course the significance for the simulation of total runoff depends strongly on the catchment (i.e., its glacier coverage). However, even if in some catchments the quantitative effect on simulated runoff would be relatively small, an adequate representation of changing glacier area is definitely desirable because it enables additional model validation (in terms of glacier simulation) and will help in the identification of the most appropriate snow/ice related model parameters being crucial for modeling alpine catchments. Below we add a table showing the model performances of the different model variants as additional information.

| NSE | GACR | No GACR | GACR-w | GACR-a |
|--------|-------|---------|--------|--------|
| Min | 0.817 | 0.818 | 0.774 | 0.801 |
| Median | 0.837 | 0.851 | 0.839 | 0.831 |
| Max | 0.855 | 0.863 | 0.854 | 0.843 |

*Mass conservation: The dh-parameterization aims at being mass conserving which is*
*crucial for hydrological modelling. In many implementations of the dh-parameterization,*
*mass conservation is a critical issue and can be violated if it is not explicitly ensured.*
*The authors should check if mass is conserved in their implementation and describe*
*their strategy to ensure mass conservation.*
This is an important comment. We can confirm that we did ensure mass conservation including a redistribution of water in model stores when some glacier elevation zones melt out and change landcover type (actually, we now realized that this technical detail needs to be mentioned in revised manuscript). We added some text to explain this more explicitly in the revised version (see also new flowchart, Figure 1).

*Different glaciers: It is unclear what happens if different (separated) glaciers are*
*present in the catchment. Can the authors' implementation of the parameterization*
*only be applied to catchments that contain one glacier? What are the limitations when*
*several glaciers are present in the catchment?*
If there are several glaciers within one (sub)catchment these are represented in a summarized way, i.e. as one glacier. From the perspective of glacier geometry change modelling this is a clear limitation as

discussed in the second paragraph of the discussion section. But the model could, in principle, be extended to include several glaciers or the model could be applied to individual glaciers separately. However, for many hydrological applications the simulation of several (small) individual glacier areas as one model glacier may be an acceptable or even practically necessary solution as the glacierized portion of the catchment is often small and similar conceptually summarized processes are used in hydrological models for other runoff generation processes in the (larger) non-glacierized part as well.

*Model calibration and validation: HBV-light is applied for an Alpine catchment for a period of >100 years. It remains unclear in the present paper how the model was calibrated and validated for this application. Some more details are necessary.*
Calibration and validation are described in detail by Stahl et al. (2017) and we added some more information on this in the revision.

Impact on runoff: see also comment above. Here, the present study using a simple and operational hydrological model could go one step further than previous studies: What is the effect of using the glacier retreat parameterization on calculated runoff? Is it possible to quantify the benefit?
This is an important aspect – also see our responses above. The effect of considering changing glacier areas on runoff differs of course depending on catchment and glacier area change. We demonstrate the effect with an example here, but more examples are provided by Stahl et al. (2017). The new routine will potentially be even more important for the simulation of future scenarios beyond 'peak water'.

*Detailed comments:*
*Page 1, line 34: Some references should be provided here*
We added four references to more complex models.

*Page 2, line 2: Actually, full hydrological models, incorporating glacier dynamics explicitly, have been published in the last years (e.g. Naz et al., 2014, HESS; Frans et al., 2016, HP). Reference to these approaches should be made, also to justify the use of strongly simplified glacier models.*
Thanks for making us aware of these studies, which we included in the revised version together with some more justification on why we choose a simpler approach.

*Page 3, line 21: ice accumulation => snow accumulation*
We changed this to Snow and ice accumulation

*page 4, line 16: A transformation time of 1-3 years is too fast. Please provide a reference and choose more realistic numbers*
We agree. What we meant was that it takes 1-3 years until snow is transformed to firn (https://nsidc.org/cryosphere/glaciers/questions/formed.html). The further transformation to ice varies with climatic conditions and can take 10 to more than 100 years. However, when simulating melt rates, we here treated the firn as ice when simulating melt rates. One can also note that this approach and even the used parameter value agree with the approach used for the snow-firn(ice) conversion by Luo et al. (2013 in Journal of Hydrology). We clarified this in the text.

*page 4, line 17-22: The description of snow redistribution is unclear and needs revision. There seems to be quite arbitrary choices in this approach and justification is required.*

We expanded the description slightly. However, this aspect is not the focus of this manuscript. Snow redistribution is a challenge on its own, which is further discussed in Freudiger et al. (WIRES Water, 2017– reference added in the manuscript).

*Page 4, line 26: "single-valued relation between glacier mass balance and glacier area". Is this really the case? This does not make sense in my opinion and also seems to be inconsistent with the argumentation in the paper. Has the word "area CHANGE" been lost? But even then, the dh-parameerization should be prescribe such a single-valued relation.*

Thanks. In the revision we deleted the word balance, i.e. relation between glacier mass and area (actually we realized that this had to be done at several places). Yes, the application of the $\Delta$h-parameterization leads to such a single-valued relation between glacier mass and glacier area in each of the different elevation zones. Section 2.1 has undergone significant revisions as a whole for clarification.

*Page 4, line 31: Here, and elsewhere. I do not like the partly very method-specific descriptions. Of course the implementation in the HBV-light model relies on a so-called "glacier profile" file. But the paper aims at providing a methodological description for implementing a glacier retreat model. So, I would avoid notions that are too specific to the authors' own model.*

We agree, but argue that it is important to clarify what the user needs to specify. Please note that we on purpose did not mention things like file names etc. We kept the term glacier profile in the revised manuscript also to refer briefly to the distribution of glacier thickness over elevation and introduced thisterm more clearly.

*Page 5, line 29: Where is h_i,old taken from? (see also general comment above)*

h_i,old is computed iteratively, we changed the notation to k+1 and k to make this clearer and added some text.

*page 6, line 35: Please provide a reference for glacier area in 2010 and a more accurate number (i.e. 1-2 digits).*

We provided digits and references. The estimates are 4.03 km$^2$ for 2010 (based on Fischer et al. 2014) and 4.57 km² for 2003 based on Paul et al. 2011

*Page 7, line 26: Where is glacier surface geometry for the year 1900 taken from?*

This is described further down (following paragraph) and we added an appendix describing this in more detail.

*Page 8, top: I suggest having a kind of data section here to better organize the input data for the example catchment*

We extended the description of the input data.

*page 8, line 22: It is not clear where the initial distribution of ice thickness around 1900 is coming from.*

See above, see appendix

*Page 8, line 29: What is done here exactly? It seems that in addition to the dh-parameterization also volume-area scaling has been used. Please describe how and why.*

*I strongly suggest to not combine volume-area scaling and the dh-
parameterization. These are separate approaches that conceptually do not go together
well*

This text does not refer to the volume-area scaling implemented in the glacier routine but to the initial conditions required for model setup. However, as this appears to be unclear, :

1)        We clarified that this (old page 8 line 29) is about the reconstruction of the initial ice thickness distribution. This text refers to the reconstruction for which we provide a more detailed explanation in the revised version (see comment above). Basically our reconstruction is mainly based on two physically-based relationships taken from Bahr et al. (1997):  i) the general volume-area scaling relation ($V = c \cdot A^{\gamma}$ with $\gamma = 1.375$) and ii) a proportionality of glacier width and the square root of glacier thickness. For clarification: While the latter relation is indeed also used in our model approach to represent the change of glacier width within a certain elevation band (as explained on page 7, eq. 7) and also applied by Huss & Hock 2015), the "classical" glacier volume-area scaling is not directly used in the modelling itself. Hence, it is not used in combination with the $\Delta h$-parameterization in the implementation of our glacier dynamic representation approach as perhaps suspected. It was solely used to derive estimates on total glacier volume for the state in the years ~1900, ~1940, and 2003. These obtained glacier volume estimates were used as data for model initialization (initial ice thickness distribution) and calibration (see observation based data in Figure 4 b /previously 3b).

2)        Regarding combining volume-area scaling with the $\Delta h$-parameterization, we argue that the original Huss approach is somewhat unrealistic as the glacier area only changes at the glacier terminus. The scaling approach allows considering the fact that a thinning glacier in some elevation zone also causes (small) decreases in area. We do not see a fundamental problem of the use of a scaling approach to allow for these (small) areal changes.

*Page 8, line 31: better 900 kg m-3*
Changed (now in the Appendix)

*page 9, line 20: Instead of using only glacier areas for model validation, the change in
glacier volume would be a much better measure to assess model performance in terms
of discharge. Such data would be available for the investigated catchment based on
Fischer et al. (2015, The Cryosphere).*

This is an interesting suggestion. However, please note that the data from Fischer et al. (2015) covers the period 1980-2010, which makes such a validation difficult since our simulation stops in 2006 due to the available input data. Therefore this option might be less suitable after all.  However, please note that Figure 4 b /previously 3b) (simulated change in glacier water equivalent) already visualizes the performance in terms of change in total glacier volume. The shown observation based change in glacier ice water equivalent values for the years 1940, 1973, and 2003, which were also used for model calibration, actually are directly converted from our estimates on glacier volume based on glacier area data for those years. We agree that for hydrological modeling an accurate simulation of glacier volume matters more than that of glacier area. That is why we used glacier volume changes (as glacier water equivalent changes) for model calibration. However, the focus of this technical note is not the hydrological simulation of changing glacier volumes per se, but how a change in volume can be translated into changes in glacier geometry and land cover type in the model. We think an additional validation using the data by Fischer et al. (2015) might be mainly informative regarding the uncertainty of the glacier volume estimates used by us (differences compared to reference data by Fischer et al.) rather than regarding the capability of the model to represent the change in glacier volume adequately.

*Page 11, line 16: Well, as the authors describe in the introduction, this approach al-*

*ready has been implemented in other hydrological models. These sentences should*
*be reformulated to better reflect this.*

We were apparently not clear what 'this approach' is referring to. Here we meant our approach (specific implementation to represent glacier dynamics in a semi-distributed hydrological model), whereas the reviewers' comment refers to the dh-parameterization. We clarified this and rephrased several sentences in the discussion section to better reflect this as suggested.

---

## Author Response (AR2)

We thank the editor and the reviewer for their continued efforts which clearly help to further clarify and improve this technical note. Below we respond (*blue italic text*) in detail to the different comments (black).

Best regards,

Jan Seibert, on behalf of all co-authors

**Editor comments:**

The reviewer has done a swift and thorough job of reviewing the manuscript and for the most part if the authors respond to these well (as they did before) then we should be able to proceed to publication. However I want a stronger clearer statement as to how this is a technical note and therefore has the appropriate novelty before this is accepted, cheers, Jim

> *We appreciate the positive evaluation of our revisions. We want to emphasize that in this technical note we describe how existing approaches can be combined in a novel way to simulate changing glacier areas in a hydrological catchment model. To date, few of the widely used hydrological catchment models allow simulating changing glacier areas inside the model. For some of these models there are plans of implementing this (e.g. PREVAH) and hence presenting a flexible approach how to do this is timely. A novel technical aspect to the general delta h approach is the use of a lookup table, which allows advancing glaciers. It should be noted, that when we started this work and discussed different options with Matthias Huss, he raised the issue that their approach would not allow increasing areas. So even if we used their existing approach we modified it in an important way that will be of interest to the community. Furthermore, the approach of a lookup table can also be used with other glacier models, for example one could run a dynamic glacier model to simulate the melt of the glacier and in this way obtain the volume-area information for the lookup table. In other words, this approach is really flexible and allows the use of more complex models if the necessary data is available (of course still with the limitation of negligible delays between mass and area changes). We now emphasize this aspect clearer in the manuscript.*

**Reviewer comments**

The authors have provided a well-formulated and complete revision of their manuscript. Most of the comments raised by the reviewers were addressed and new analysis has been added to the paper. The article presents the method (glacier retreat scheme for lumped hydrological models) in a clear and reproducible way and will be helpful to the community. Nevertheless, there are some (mostly minor) issues with the present version of the article that should be addressed before final acceptance.

*Thanks for this kind assessment of our revisions. We would like to stress that the approach can also simulate glacier advance and that the model should be considered semi-distributed due to its use of HRUs by elevation and aspect classes.*

- Page 5, line 15: The reasoning is somewhat unclear here. Obviously, the authors have a complete snow redistribution model available (is this described in Freudiger et al, 2017? – the text seems to provide this hint) but decide not to use it and go for an extremely simple approach. I would avoid mentioning the "other" approach for removing "snow towers" and just describe what has been done.

*Actually, in the review paper of Freudiger et al 2017 the challenges of modeling snow restribution are described but no 'best' or 'complete' snow redistribution model is presented.. We clarified in the text what we meant in a general way and what our approach was.*

- Page 5, line 19: Please shortly explain how the parameters were chosen and if there was any possibility to validate them.

*We added the values and a short motivation of these choices. Obviously, the whole issue of snow redistribution and its representation in a simple catchment model would motivate a study on its own, but this was not the focus here.*

- Page 5, line 25: The "lookup table" seems to be one of the major modifications in comparison to the original dh-parameterisation. It is mentioned for the first time here but unfortunately it remains highly unclear what this table (1) actually is and does, and (2) how it is derived. Regarding (1): This is clarified later in the paper (with the aid of the figure). However, it should be outlined here. Regarding (2): The formulation calls for better explanation. "… a lookup table can be derived from any glaciological model". So, another model (a glaciological model) is needed to generate this lookup table? Which model was used by the authors? Apparently, this might quite drastically limit the applicability of the authors' proposed model implementation as not all hydrologists might have a glaciological model at hand to produce this lookup table. Maybe it is just a misunderstanding that can be clarified with a better formulation?

*1) We are glad that the lookup table approach became clear now and changed the text to better describe already here what the lookup table is.*

*2) This is a misunderstanding. What we meant to say is that other models could also be used to create the lookup table instead of the Δh-parameterization method. We changed the text to clarify this, in particular we moved this point entirely to the discussion.*

- Page 10, line 28: The area change 1900-2006 is correctly calculated by the full model. However, the two individual periods 1900-1940, and 1940-1973 are completely off. This should also be mentioned. I suggest providing the results for relative area change (observed / simulated) for all four experiments in a table. This would better allow tracking how well the model implementations work.

*We compiled this information in the form of tables below.*

*For the text of the technical note, however, we do not find this information so helpful. The basic information is given in the Figures and we feel that providing the exact values rather confuses than helps. Please also note that we do not want to pretend that the new model routine is fully validated quantitatively.*

Table: Absolute area [km²]

| Reference year | Glacier area [km²] | | | | | | | |
|---|---|---|---|---|---|---|---|---|
| | Ref | GACR | | GACR-a | | GACR-w | | No GACR |
| | | Min | Max | Min | Max | Min | Max | |
| 1901 | 6.23 | 5.92 | 5.98 | 5.92 | 5.98 | 6.21 | 6.23 | 6.23 |
| 1940 | 6.16 | 5.65 | 5.65 | 5.54 | 5.71 | 6.00 | 6.14 | 6.23 |
| 1973 | 5.20 | 5.05 | 5.15 | 4.74 | 4.84 | 5.71 | 5.71 | 6.23 |
| 2003 | 4.54 | 4.45 | 4.55 | 4.51 | 4.55 | 5.34 | 5.42 | 6.23 |

Table: Catchment glacier coverage [%] another option would be relative change compared to initial ref. area

| Reference year | Catchment glacier coverage [%] | | | | | | | |
|---|---|---|---|---|---|---|---|---|
| | Ref | GACR | | GACR-a | | GACR-w | | No GACR |
| | | Min | Max | Min | Max | Min | Max | |
| 1901 | 30.1 | 28.6 | 28.9 | 28.6 | 28.9 | 30.0 | 30.1 | 30.1 |
| 1940 | 29.8 | 27.3 | 27.3 | 26.8 | 27.6 | 29.0 | 29.7 | 30.1 |
| 1973 | 25.1 | 24.4 | 24.9 | 22.9 | 23.4 | 27.6 | 27.6 | 30.1 |
| 2003 | 21.9 | 21.5 | 22 | 21.8 | 22 | 25.8 | 26.2 | 30.1 |

Table: Relative difference of simulated glacier area compared to reference glacier data

| Reference year | Ref. data [km²] | Rel. difference of simulated glacier area compared to ref. data [%] | | | | | | |
|---|---|---|---|---|---|---|---|---|
| | | GACR | | GACR-a | | GACR-w | | No GACR |
| | | Min | Max | Min | Max | Min | Max | |
| 1901 | 30.1 | - 5.0 | - 4.0 | - 5.0 | - 4.0 | - 0.4 | - 0.1 | 0.0 |
| 1940 | 29.8 | - 8.3 | - 8.3 | -10.0 | - 7.3 | - 2.6 | - 0.3 | + 1.1 |
| 1973 | 25.1 | - 3.0 | - 1.0 | - 8.9 | - 6.9 | + 9.8 | + 9.8 | +19.7 |
| 2003 | 21.9 | - 2.0 | + 0.2 | - 0.7 | + 0.2 | +17.5 | +19.7 | +37.1 |

- Page 11, line 9: A general comment on "glacier advance". The authors directly link increases in glacier mass to advances of the glacier front. They do not consider a temporal delay. This strongly contradicts

observations: Several years or even decades of mass gain are required (depending on glacier size and shape) until ice flow dynamics have changed in a way to make the glacier snout advance. This is completely neglected in the presented approach. I fully understand that this is not feasible, and probably also not necessary (!), for such a model, but this effect should be critically discussed by the authors and formulations should be adapted throughout the paper to not imply equality of glacier mass gain and glacier advance.

*We fully agree and clarified that we make the assumption that delays between glacier mass and area changes can be neglected and that glacier retreat and glacier advance follow the same (but reverse) pattern. At the same time, as the reviewer noticed, for catchment models focusing on ice melt runoff the more complex details of delays in the glacier responses are not feasible to implement, and probably also of secondary importance. We addressed this more clearly in the discussion section.*

- Page 12, line 22: This statement should be corrected. Modern remote sensing data provide elevation changes for virtually all glaciers in the world (see e.g. Brun et al., 2017, NGEO). Glacier-specific elevation changes are also available for all glaciers in Switzerland, for example (Fischer et al., 2015, The Cryosphere).

*We have to agree with the reviewer that data often would be available to established individual local Δh-parametrizations. What we meant to say was that it is in practice much easier to use established Δh-parametrizations than to establish specific relationships between mass balance changes and glacier thickness distributions for future conditions of individual glaciers. We changed the text to clarify this.*

- Figure 1: Nice! However, I strongly recommend enlarging text size (and reduce unfilled white spaces) to make the figure readable more easily.

*Thanks, we enlarged the text size*

- Figure 3c: Maybe show relative instead of absolute errors? This would be more informative. Why is there a difference for the 1900 surface?

*We considered relative errors, however, deemed those as less informative as the values get large for small glacier areas in lower elevation zones even when errors are not that big (compared to other elevation zones)*

*The difference in the values for 1900 arise from differences during the warm-up period of three years prior to 1900, during which the glacier already decreased. We clarified this in figure 4 and the figure caption.*

- Figure 4a: Why do not all runs start with the same area? Also 1900 should have a dot for an observed area. The dots should have error bars (!) and might be linked with lines to make the figure clearer.

*The difference in the values for 1900 arise from differences during the warm-up period of three years prior to 1901, during which the glacier already decreased. We clarified this in the text and*

*added the warm-up period in Figure 4 as well as the initial glacier area taken from the historical maps (Freudiger et al., ESSDD)*

*We agree that the shown glacier area values are far from certain and added the source of the individual values and error bars, which correspond to an accuracy of +/- 5 %. It is based on the accuracy given in the references Freudiger et al. (ESSDD), Paul et al. (2011), and Fischer et al. (2014). In the references for the glacier inventory 1973 there isn't any uncertainty specified, but we assume a similar accuracy of +/- 5% as for the inventories from 2003 and 2010. For the two early glacier area values, it should be noted that an inaccuracy of +/- 5 % has been attributed to the digitization of the historical maps by Freudiger et al. (ESSDD, in review), while an additional considerable uncertainty of the glacier outlines or the historical maps itself needs to be assumed, yet remains very difficult to ascertain. Hence, for those values +/- 5% should be regarded as lower bound value. Additionally horizontal error bars, i.e. temporal uncertainty, are shown in the Figure for the two early glacier area values from the historical maps. These horizontal error bars correspond to the period between the release dates of the 2 individual map sheets that cover the Alpbach catchment, i.e. 1894 and 1899 for the first value used as initial glacier area in the simulations and 1933 and 1942 for the latter value. Probably even a period extending further to the past could have been indicated, since the underlying survey had probably taken place before the release year of the individual map sheets.*

*We didn't add lines linking the reference glacier area point values, because of the long time differences between the points, the different data sources.*

- Figure 4b: This is not observed volume! A well-constrained bedrock topography from the dataset of M. Huss is available for 1973 and 2010 but the volumes for 1940 and 1900 are derived from volume-area scaling, i.e. an extremely simple and highly uncertain model. So, it cannot be used for model validation! As already suggested in my last review, there are actual measurements of ice volume change for the respective glaciers (Fischer et al., 2015). They do not refer to the entire study period but nevertheless would allow validation over several decades.

*We agree that the shown volume estimates should better not be termed observation-based. We changed the figure legend and used different symbols for the values 1973 & 2010 (based on data from Matthias Huss) and the other values. We are aware of the high uncertainties especially in the 1900/40 source data as well as in the volume estimates for the years 1901, 1940, 2003 related to the volume–area scaling but argue that these data are still better than having no information at all. Actually, these data are not used for model validation but for model initialization (initial glacier volume) and calibration. Since the model has been calibrated with the shown calibration, it cannot be used for model validation. It was not our intention to validate the glacier mass balance here, but to present the newly introduced glacier area change approach with an example dataset and to demonstrate the effect of the different tested variants on e.g. the simulated glacier mass balance. It is clear that the inconsistency or inhomogeneity due to the combination of these different glacier datasets and volumes based on simple volume–area scaling and the thickness data based on more appropriate data and the approach presented in Huss & Farinotti (2012) is problematic if one aims at getting most accurate values for individual glaciers. For our example application starting as*

*early as 1901 more appropriate data have not been available. We do not consider this a crucial issue for the demonstration of the glacier area change routine, but we added in the some text critical remarks on the volume–area scaling.*

*The challenge with the Fischer et al 2015 ice thickness change data is, as mentioned in our previous response, that our simulations end in 2006 (due to the data availability of the HYRAS data set) As said above, for us validation of the simulated mass balance (volume changes) has not been the focus of this technical note. However, we now added the geodethic ice thickness change from 1981– 2010 from the WGMS / Fischer et al. (2015) in the figure.*

- Figure 4c: Please change y-axis label. Glacier runoff is normally interpreted as the water leaving the glacier. These water volumes are much higher as they also comprise seasonal snow melt on the glacier surfaces. The authors show cumulative glacier mass loss here, and not runoff. The legend of Figure 4 should be positioned more prominently – I first didn't find it. Furthermore, I would suggest to not use model abbreviations but more intuitive names.

*We agree that the term glacier runoff can be confusing as it is used differently by different authors, we used it as runoff generated by ice melt only (see also Weiler et al., early view). To avoid any confusion we changed the term to 'Glacial ice melt runoff'. We clarified in the figure caption that this term refers to melted ice only (i.e., not snowmelt on the glacier) which is tracked through the model (Weiler et al., early view).*
*The individual graphs of the figures were arranges closer to each other and the legend was enlarged, we think this should be prominent enough !? For consistency, we would like to keep the abbreviations for the model variants, which we define in Sections 2.2 and use in the text, when we refer to the variants. We think clear synonyms for the variants would become too lengthy for the figure legend and would make the text hard to read.*

- Page 26, line 18: Combining volume-area scaling with the ice thickness distribution data set derived by a different methodology seems to be inconsistent, leading to a non-continuous evolution of glacier volume. In any case, the derived volumes for 1900 and 1940 using this method will not be suitable to validate the model regarding glacier mass change (Fig. 4b)! A better discussion is needed.

*We are aware of the uncertainties in the 1900/40 data but would argue that uncertain data is better than no data at all.*

See also comments re Fig. 4 b above. In the last paragraph of the appendix we already pointed to the considerable uncertainties to be taken into account. In the revised version, we added to this discussion that the combination of volume-area scaling with the ice thickness distribution data set derived by a different methodology results in additional uncertainty and that, if feasible (required date available), it would of course be better to use consistent methodolopgical approaches for all glacier data estimates.

*Weiler M, Seibert J, Stahl K., early view. Magic components—why quantifying rain, snowmelt, and icemelt in river discharge is not easy. Hydrological Processes.*
[https://doi.org/10.1002/hyp.11361](https://doi.org/10.1002/hyp.11361)